# PRIM: Cooperative Dynamic Token Compression for Efficient Large Multimodal Models

**Song Li**[1]   **Yongping Xiong**[1]

## Abstract

Large multimodal models (LMMs) have shown strong capabilities in audio-visual understanding by jointly reasoning over visual, auditory, and linguistic inputs. However, processing long-form audio-visual content often requires a large number of tokens, leading to substantial computational and memory overhead during inference. Existing efficiency-oriented methods typically apply uniform compression or pruning strategies, which overlook modality-specific characteristics and instruction-dependent reasoning behaviors in multimodal models. In this work, we present **PRIM**, an inference framework for efficient multimodal reasoning that systematically compresses audio-visual representations based on attention dynamics and instruction relevance. Motivated by an attention-based analysis revealing modality imbalance and layer-wise redundancy in LMMs, PRIM introduces a cooperative compression pipeline that spans both multimodal encoders and the language model. Specifically, PRIM performs early text-conditioned audio-visual fusion to externalize cross-modal interactions, applies attention-guided and frequency-aware strategies to remove redundant audio and video tokens, and further adapts token retention inside the language model according to task demands. Extensive experiments on multiple audio-visual benchmark datasets demonstrate that PRIM consistently achieves stable and superior efficiency–accuracy trade-offs across diverse tasks. These results demonstrate that PRIM, a multimodal cooperative compression approach, provides an effective pathway toward scalable and efficient audio-visual reasoning.

[1]State Key Laboratory of Networking and Switching Technology, Beijing University of Posts and Telecommunications, Beijing, China. Correspondence to: Yongping Xiong <ypxiong@bupt.edu.cn>.

*Proceedings of the 43rd International Conference on Machine Learning*, Seoul, South Korea. PMLR 306, 2026. Copyright 2026 by the author(s).

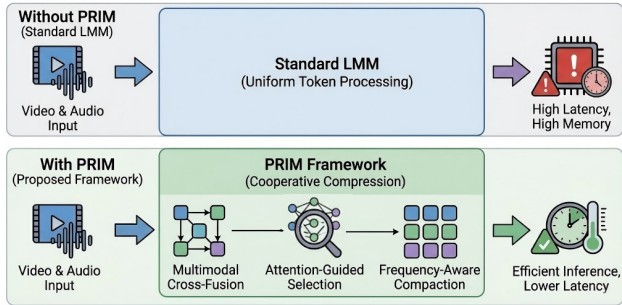

*Figure 1.* **Illustration of the computational bottleneck in standard LMMs versus the efficient solution offered by PRIM.** The top pipeline shows how uniform processing of audiovisual tokens leads to prohibitive computational costs. The bottom pipeline demonstrates how PRIM's cooperative compression modules significantly reduce redundancy to enable scalable, efficient inference.

## 1. Introduction

Large Multimodal Models (LMMs) have recently demonstrated remarkable progress in audio-visual understanding by jointly modeling visual, auditory, and linguistic signals within a unified Transformer-based framework (Alayrac et al., 2022; Liu et al., 2023). By encoding videos and audio streams into long sequences of tokens and conditioning generation on natural language instructions, modern LMMs are capable of performing complex reasoning tasks such as event localization, cross-modal grounding, and temporal inference (Sun et al., 2023; Zhang et al., 2023). However, this expressive power comes at a significant computational cost: a single video with accompanying audio can easily produce tens of thousands of tokens, leading to prohibitive memory consumption and inference latency (Shao et al., 2025). This issue becomes particularly severe for long-form videos and real-world deployment scenarios, where efficiency is a primary concern.

Existing efforts to improve the efficiency of LMMs primarily focus on architectural simplifications, low-rank adaptations, or uniform token pruning strategies inherited from vision-only models (Rao et al., 2021; Ryoo et al., 2021). While effective to some extent, these approaches often overlook a fundamental characteristic of multimodal reasoning: not all modalities contribute equally, and not all tokens are equally informative across time, frequency, or instruction

context (Wu et al., 2025). Treating audio, video, and language tokens uniformly can therefore potentially result in either excessive information loss or limited computational savings.

In this work, we argue that efficient multimodal inference should be guided by how LMMs actually attend to multimodal information during reasoning. By systematically analyzing attention distributions across layers and modalities, we uncover two key observations. First, multimodal interactions are most prominent in the early layers of the language model, while deeper layers primarily operate on more compact, instruction-aligned representations. Second, audio tokens consistently receive higher attention weights than video tokens across layers, suggesting that audio carries denser semantic cues, whereas video tokens exhibit substantial spatial and temporal redundancy. These observations motivate a principled, modality-aware compression strategy that dynamically adapts to both input structure and task requirements.

Based on the above insights, we propose **PRIM**—an efficient multimodal compression framework specifically designed for multimodal audio-visual understanding, as illustrated in Figure 1. Instead of applying compression solely within the language model, PRIM introduces a cooperative design that combines early multimodal fusion with specialized, signal-aware compression strategies. First, we introduce a multimodal cross-fusion module that explicitly performs text-conditioned audio-visual fusion before the language model, effectively shifting essential cross-modal interactions out of the Large Language Model (LLM) and enabling substantial token reduction without degrading semantic alignment. Second, we propose an attention-guided selection strategy that dynamically identifies informative audio tokens within each temporal window and uses audio saliency to adaptively control video token retention, ensuring that semantically critical events are preserved. Third, we exploit the frequency-domain characteristics of visual representations and introduce a frequency-aware compaction mechanism that removes spatial and temporal redundancy in video tokens through lightweight, parameter-free transformations. Finally, to accommodate varying task complexity and instruction specificity, we design an instruction-aware adaptation strategy that performs adaptive token pruning inside the LLM by selectively leveraging high-quality multimodal attention heads.

Together, these components form a cooperative and interpretable compression pipeline that significantly reduces computational overhead while preserving—and in many cases improving—model performance. Extensive experiments on diverse audio-visual benchmarks demonstrate that PRIM consistently outperforms strong baselines under comparable or lower token budgets, particularly in long-video

and reasoning-intensive settings. Our results suggest that compression guided by multimodal attention and instruction relevance is not merely an efficiency optimization, but a pathway to more robust multimodal reasoning. Our main contributions are summarized as follows:

- We propose **PRIM**, a cooperatively compressed efficient inference framework for Large Multimodal Models, which systematically and dynamically compresses audio-visual representations based on attention features and instruction relevance to enable scalable inference.

- We introduce a multimodal cross-fusion module that enables early, text-conditioned fusion of audio and visual information, significantly reducing reliance on internal LLM interactions for multimodal alignment.

- We propose attention-guided and frequency-aware compression strategies that dynamically remove redundant audio and video tokens while preserving essential semantic structure for downstream reasoning.

- We design an instruction-aware adaptation mechanism that performs prompt-dependent token pruning inside the LLM without additional training.

- We conduct extensive experiments on multiple audio-visual benchmark datasets, demonstrating that PRIM consistently exhibits stable and superior efficiency–accuracy trade-offs across diverse tasks.

## 2. Preliminaries

This section introduces the overall architecture of large multimodal models and provides an analysis based on token-level attention distributions.

### 2.1. General Architecture of Large Multimodal Models

Large multimodal models are advanced multimodal architectures that integrate visual, audio, and language processing capabilities. Their core objective is to unify multimodal inputs with user instructions to construct a comprehensive audio-visual understanding system. Specifically, such systems rely on Transformer-based encoders to process visual and audio inputs, while an LLM is employed to handle textual information, enabling the model to generate a language response $X_o$ conditioned on a given language instruction $X_q$, visual input $X_v$, and audio input $X_a$. These models typically adopt a modular design, consisting of a visual encoder, an audio encoder, a projection module, and an LLM backbone. The visual encoder is responsible for extracting visual features, the audio encoder captures audio representations, the projection module maps visual and audio embeddings into the semantic space of language tokens, and the LLM performs multimodal reasoning and text generation.

Specifically, models in the Qwen family, such as the Qwen-Omni series (Team, 2024), typically adopt pretrained Transformer-based backbones from the Qwen series as visual and audio encoders, and employ a two-layer MLP as the projection module. The overall input format of the model is structured as follows:

> *(system prompt)*
> USER: `<video> <audio>` *(user instruction)*
> ASSISTANT:

The video placeholder token `<video>` and the audio placeholder token `<audio>` are placed immediately after the system prompt, followed by the user instruction. We first decompose the video into individual frame clips $X_v \in \mathbb{R}^{T \times H \times W \times 3}$ and extract audio segments $X_a$ at a fixed sampling rate, where $T$ denotes the number of sampled frames. The visual encoder $g_v$ and the audio encoder $g_a$ transform the raw video frames and audio segments into higher-level, sequential token embedding representations, respectively:

$$H^v = g_v(X_v), \quad H^a = g_a(X_a) \tag{1}$$

where $H^v \in \mathbb{R}^{N_v \times D_v}$ and $H^a \in \mathbb{R}^{N_a \times D_a}$ denote the visual and audio token representations, respectively, where $N_v$ and $N_a$ are the numbers of visual and audio tokens, and $D_v$ and $D_a$ indicate the corresponding output dimensions. The projection module maps these multimodal tokens into the embedding space of the large language model, enabling effective processing of multimodal inputs and yielding a formalized sequence representation:

$$\langle H_1^q \dots H_k^q, \ H_1^v \dots H_{l_v}^v, \ H_1^a \dots H_{l_a}^a, \ H_{k+1}^q \dots H_{l_q}^q \rangle \tag{2}$$

where $l_v$, $l_a$, and $l_q$ denote the lengths of the visual, audio, and textual token sequences, respectively. In large multimodal models, the system prompt (i.e., $H_1^q \dots H_k^q$) is placed before the visual and audio tokens, while the user instruction (i.e., $H_{k+1}^q \dots H_{l_q}^q$) is positioned after the visual and audio tokens. Such a token sequence, in which the textual tokens are interleaved with visual and audio tokens, forms the input context of the large language model. The resulting sequence is then fed into the LLM to generate the final response $Y$.

In practice, the concatenation of audio and video tokens is performed using fixed-duration temporal windows. The audio and video streams are segmented into multiple windows of equal length. Within each window, temporally aligned multimodal tokens are synchronized and concatenated into cross-modal blocks. These blocks are then concatenated in chronological order to form a long token sequence, which is subsequently fed into the large language model. The LLM generates responses by jointly aligning video, audio, and textual representations. In typical scenarios, a single video with accompanying audio can produce tens of thousands of

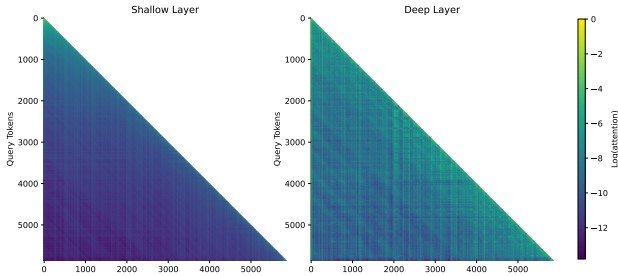

*Figure 2.* **Attention distribution evolution across different layers during the LLM inference process.** The horizontal axis represents key positions, while the vertical axis represents query positions. We visualize the attention maps of the first and the last Transformer layers. In the shallow layers, attention is broadly distributed across tokens, reflecting early-stage multimodal token mixing. In contrast, in the deeper layers, attention becomes more localized and sparse, indicating a shift toward semantic abstraction and a reduced reliance on raw token-level interactions.

tokens after encoding. Such extremely long sequences incur substantial computational and memory overhead, severely hindering efficient deployment and real-time inference.

## 2.2. Token Analysis

To preserve the model's understanding capability while compressing tokens, we investigate how large multimodal models interpret visual and audio tokens in order to guide effective token compression strategies. Specifically, we analyze the role of these tokens in LMMs from the perspective of the attention mechanism (Zheng et al., 2024). To characterize the redundancy and attention patterns of different modality tokens during the reasoning process, we visualize the attention distributions.

Representative self-attention maps from a shallow layer and a deep layer are presented in Figure 2. In shallow layers, attention is broadly distributed across tokens, where visual and audio tokens receive substantial attention in early layers, indicating extensive token-level mixing and relatively uniform aggregation of information across different tokens. This behavior suggests that visual and audio tokens play a central role in early layers. In contrast, deep layers exhibit more concentrated and structured attention patterns, with a clear decreasing trend in attention allocated to video and audio tokens. Notably, significantly higher attention weights are assigned to only a small subset of tokens. This shift reflects that while nearly all visual and audio tokens receive broad attention in early layers, later layers increasingly focus on fused tokens that interact with instruction-related information. Such a contrast highlights the pronounced evolution of attention across layers and supports the observation that deeper layers operate on more compact and abstract representations rather than raw token-level features. Motivated by this insight, we introduce a multimodal cross-fusion mod-

ule that moves the fusion process from the early layers of the large language model to a stage prior to the LLM. By doing so, we can substantially reduce the number of visual tokens fed into the LLM without sacrificing performance, thereby improving inference efficiency.

The analysis of cross-modal attention ratios across LLM layers, aiming to reveal the differing contributions of each modality during the reasoning process, is illustrated in Figure 3. By closely examining the distribution of inter-token interactions, we observe that audio tokens consistently receive higher attention weights than video tokens across all layers. This indicates a clear dominance of audio tokens over video tokens, suggesting that audio carries denser semantic information and plays a primary role in multimodal reasoning. In contrast, large regions of video tokens exhibit significantly lower attention scores, implying substantial structural redundancy within the video modality. Based on this key observation, we design a dynamic compression strategy that performs token pruning independently within each temporal window. Specifically, modality-specific tokens are compressed separately, with the retention ratio of audio tokens serving as the dominant factor to dynamically adjust the video pruning ratio for each temporal window, while ensuring that the video compression rate is always higher than that of audio. This mechanism leverages the semantic guidance provided by audio to effectively remove redundant video tokens, thereby significantly reducing computational and memory costs while maintaining performance on downstream tasks.

## 3. The Proposed Method: PRIM

In this section, we present the core contribution of this work, namely the **PRIM** method, and further detail the corresponding token compression strategies. PRIM is a multimodal large-model compression framework specifically tailored for audio-visual understanding tasks. As illustrated in Figure 4, the framework first introduces a multimodal cross-fusion module to enhance inter-modal interactions. It then proposes efficient, modality-specific compression strategies for the audio, video, and large language model components, respectively, to further achieve efficient inference optimization.

### 3.1. Multimodal Cross Fusion

Given the video and audio inputs $X_v$ and $X_a$, we employ pretrained visual and audio encoders to extract visual and audio features, respectively, which are then mapped into the token embedding space via a projection layer to produce the corresponding visual and audio tokens $H^v$ and $H^a$. For the language instruction $X_q$, textual token representations $H^q$ are generated using the embedding layer of the large language model. Since token compression inevitably leads

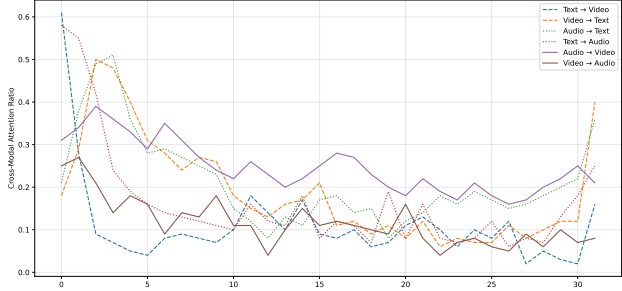

*Figure 3.* **Evolution of cross-modal attention across LLM layers.** We observe a clear pattern of *information migration*: cross-modal attention is higher in the early layers, reflecting active information exchange, while it gradually diminishes in deeper layers, indicating a transition toward fused representations conditioned on text. Notably, audio tokens consistently receive higher attention weights than video tokens across all layers.

to partial information loss, we introduce a multimodal cross-fusion module before the LLM backbone, allowing textual tokens to incorporate relevant information from all visual and audio tokens at an early stage. Based on our prior observations that such fusion implicitly occurs in the early layers of the LLM, we explicitly perform interactive attention computation using a multimodal cross-fusion module. The module $f(\cdot)$ is composed of multiple Transformer blocks (Vaswani et al., 2017), each sharing the same architecture and hyperparameters as the LLM backbone. Specifically, the visual tokens $H^v$ and audio tokens $H^a$ are concatenated with the textual tokens $H^q$ and fed into the cross-fusion module, after which the outputs corresponding to the textual tokens are extracted as the fused token representations, which can be formulated as:

$$\hat{H}^{qv} = f(\text{Concat}(H^v, H^q))\,[-l_q :]$$
$$\hat{H}^{qa} = f(\text{Concat}(H^a, H^q))\,[-l_q :] \qquad (3)$$

where $\hat{H}^{qv}$ and $\hat{H}^{qa}$ denote the fused visual and audio representations that incorporate the language instruction, respectively, and $l_q$ represents the number of textual tokens. For the fused tokens, we typically apply a pooling operation to aggregate the visual and audio fusion tokens across all frames, thereby yielding the corresponding fused token representations. Subsequently, these fused tokens $\hat{H}^{qv}$ and $\hat{H}^{qa}$ are concatenated and jointly fed, together with the compressed tokens $\hat{H}^v$ and $\hat{H}^a$, into the large language model to generate the response. In this way, the number of tokens input to the language model is substantially reduced, leading to a significant improvement in overall computational efficiency.

### 3.2. Attention-Guided Selection

For audio and video token compression, we adopt a window-wise processing strategy, in which audio and video tokens

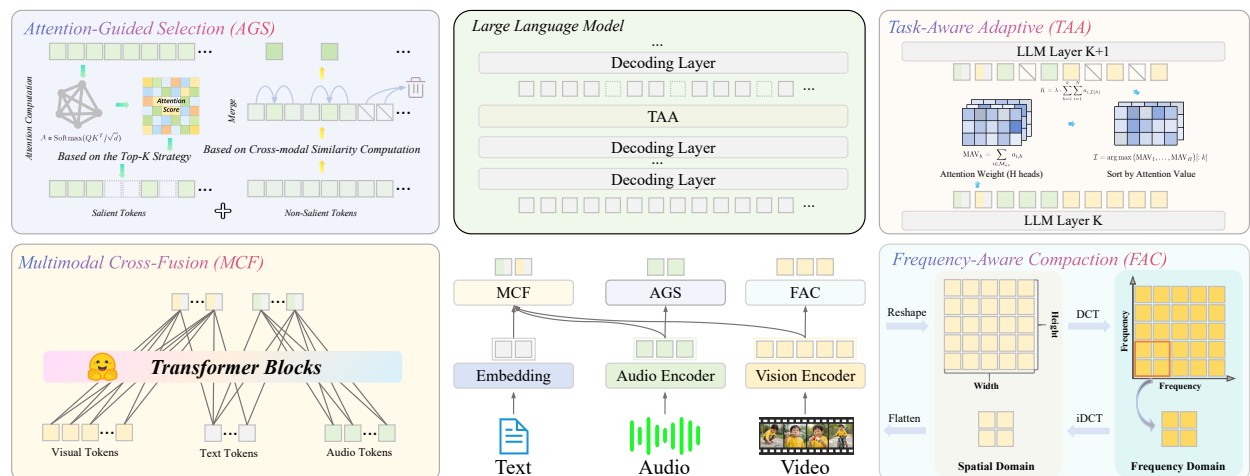

*Figure 4.* **The overall framework of the proposed PRIM.** The model first encodes text, audio, and video inputs into modality-specific token representations. PRIM then performs multimodal cross-fusion (MCF), attention-guided selection (AGS), and frequency-aware compaction (FAC) to remove redundant tokens while preserving salient cross-modal information. The compressed tokens are subsequently fed into a large language model equipped with a task-aware adaptive module (TAA) to generate the final response.

are pruned independently within each temporal window. Specifically, for audio processing, we assume that the $t$-th temporal window contains $n_a$ audio tokens and $n_v$ video tokens, whose projected embeddings are denoted as $H_{t,i}^a$ and $H_{t,j}^v$, respectively. We select informative audio tokens based on the attention distribution produced by the audio encoder, where the attention matrix from the final layer of the audio encoder $g_a$ is used to quantify the importance of individual audio tokens:

$$A = \text{Softmax}\left(\frac{QK^\top}{\sqrt{d}}\right) \in \mathbb{R}^{B \times N_a \times N_a} \quad (4)$$

where $Q, K \in \mathbb{R}^{N_a \times d}$ denote the query and key matrices of the audio tokens, respectively. $B$ denotes the batch size and $d$ is the feature dimension. We quantify the importance of each audio token by computing its average attention received from all other audio tokens, with tokens assigned higher priority if they exhibit larger average attention weights. We then adopt a top-$k$ strategy to select the audio features with the highest attention scores (corresponding to the top $S\%$) as representative and information-dense tokens, while the remaining tokens are regarded as non-salient.

Considering that non-salient tokens still carry informative cues, we integrate these non-salient tokens to preserve contextual semantics while retaining critical information. Specifically, the proposed method first uniformly samples representative tokens from the non-salient tokens within each temporal window. To maintain semantic consistency between audio and visual representations, we further evaluate the interaction similarity between audio and video tokens, thereby selecting key tokens from the candidate set:

$$Sim_{\text{cross}} = \tilde{H}^a(\tilde{H}^v)^\top, \quad Sim_{ij} = \tilde{h}_i^{a\top} \tilde{h}_j^v \in [-1, 1] \quad (5)$$

where $\tilde{H}^a$ and $\tilde{H}^v$ denote the normalized audio token sequence and the normalized video token sequence, respectively:

$$\tilde{H} = \text{Diag}\left(\sqrt{\text{diag}(HH^\top)} + \epsilon\right)^{-1} H, \quad \epsilon = 10^{-6} \quad (6)$$

Here, we select the top-$\delta$ candidate audio tokens that are most relevant to the corresponding video segment and aggregate them into representative tokens, where $\delta$ denotes the number of tokens merged into each representative token. The remaining non-salient tokens are then discarded to achieve deep compression of audio tokens.

After audio token compression, we compute an audio retention score for each temporal window, denoted as $P_a(i) \in [0, 1]$, where $i$ indexes the temporal groups. Windows with higher retention scores exhibit clearer event boundaries and higher information density. We set the video token compression ratio to $\rho_v$ and adopt a dynamic video token compression strategy: windows with higher scores are processed conservatively, while those with lower scores undergo more aggressive compression. This yields the initial adaptive ratio $\rho'_v(i)$:

$$\rho'_v(i) = \rho_{\max} - (\rho_{\max} - \rho_{\min}) \cdot P_a(i) \quad (7)$$

where $\rho_{\max}$ and $\rho_{\min}$ denote the upper and lower bounds of the pruning ratio, respectively, which are introduced to prevent excessive pruning. These initial ratios $\rho'_v(i)$ are subsequently normalized in a global manner:

$$\rho_v(i) = \frac{\rho_v \sum_j N_j}{\sum_j \rho'_v(j) N_j} \cdot \rho'_v(i) \quad (8)$$

where $N_j$ represents the number of video tokens in the $j$-th temporal window, and $\rho_v(i)$ denotes the actual video

compression ratio applied to each temporal window. We leverage $\rho_v(i)$ to guide the compression of video tokens.

### 3.3. Frequency-Aware Compaction

For video inputs, we observe that the visual features produced by the visual encoder exhibit a pronounced energy concentration in low-frequency components. Motivated by this observation, we propose a simple yet efficient approach to compress visual representations in the frequency domain.

Specifically, we apply a two-dimensional Discrete Cosine Transform (DCT) to the image features output by the visual encoder. We introduce a parameter-free module, termed the Frequency Feature Compressor (FFC), immediately after the visual encoder. This module contains no trainable parameters and can substantially reduce the number of visual tokens. We set the minimal processing unit to four frames and apply the proposed FFC module to each frame of the video independently. As a result, the number of visual tokens is reduced from $N^2$ to $C^2$, where the compression ratio is controlled by the previously computed $\rho_v(i)$.

$$H_c^v = FFC\text{-}Module(H^v) \tag{9}$$

Specifically, within the FFC module, the original visual features are first reshaped into a grid structure:

$$G^v = Reshape(H^v) \tag{10}$$

where $G^v \in \mathbb{R}^{N \times N \times h^v}$ and $h^v$ is the output dimension of the vision encoder. Subsequently, a discrete cosine transform is applied along the two spatial dimensions (both of size $N$) to obtain the frequency-domain representation $\hat{F}^v$. This process can be formally expressed as:

$$\hat{F}_{m,n,:}^v = \alpha_m \alpha_n \sum_{p=0}^{N-1} \sum_{q=0}^{N-1} G_{p,q,:}^v \cdot \phi_N(m,p) \cdot \phi_N(n,q) \tag{11}$$

where $\hat{F}^v \in \mathbb{R}^{N \times N \times h^v}$, $\phi.(\cdot,\cdot)$ and $\alpha_m$ are defined as:

$$\phi_N(x,y) = \cos\left[\frac{\pi}{N}x\left(y + \frac{1}{2}\right)\right] \tag{12}$$

$$\alpha_m = \begin{cases} \sqrt{\frac{1}{N}}, & \text{if } m = 0, \\ \sqrt{\frac{2}{N}}, & \text{otherwise.} \end{cases} \tag{13}$$

Since the energy is primarily concentrated in the low-frequency components, it is therefore necessary to prune the corresponding high-frequency components:

$$F^v = \hat{F}^v[0:C, 0:C, :] \tag{14}$$

where $\hat{F}^v \in \mathbb{R}^{C \times C \times h^v}$, $C^2$ denotes the number of retained visual tokens.

Finally, we apply the inverse discrete cosine transform (iDCT) to reconstruct spatial features from the frequency-domain tokens:

$$T_{i,j,:}^v = \sum_{p=0}^{C-1} \sum_{q=0}^{C-1} \alpha_p \alpha_q \cdot F_{p,q,:}^v \cdot \phi_C(p,i) \cdot \phi_C(q,j) \tag{15}$$

where $T^v \in \mathbb{R}^{C \times C \times h^v}$ denotes the intermediate visual representation, which is subsequently flattened to obtain compressed visual features:

$$H_c^v = \text{Flatten}(T^v) \tag{16}$$

where $H_c^v \in \mathbb{R}^{C^2 \times h^v}$. Meanwhile, within the same temporal window, we compute the cosine similarity between tokens at corresponding spatial locations across adjacent frames:

$$Sim_{\text{t}} = \cos(\theta) = \frac{h_i^v \cdot h_j^v}{\|h_i^v\| \|h_j^v\|} \tag{17}$$

We then employ the $Sim_{\text{t}}$ algorithm to evaluate temporal redundancy and perform token pruning on highly similar frames, thereby preserving salient visual tokens while discarding redundant ones.

### 3.4. Task-Aware Adaptive Pruning

Relying solely on audio and video information for token pruning is insufficient to eliminate redundancy that is specific to a given textual prompt, nor can it satisfy the varying token budget requirements of instances with different levels of complexity. For example, high-resolution tasks typically require more tokens than simpler tasks. To address this, we incorporate the textual prompt into the LLM decoding stage to perform adaptive pruning of redundant tokens. This approach is training-free and dynamically adjusts the sparsity ratio according to the complexity of the input instance.

We identify redundant tokens by computing the attention value $a_i$ between each token $z_i$ and the final instruction token $t$. We observe that not all attention heads are equally informative for accurately identifying redundant tokens; in other words, not all self-attention heads in large language models are strongly correlated with correctly mapping textual tokens to visual and audio tokens.

We introduce the Multimodal Attention Value (MAV) to measure the quality of each attention head, which is defined as the magnitude of the attention between the textual token $t$ and all visual and auditory tokens. For the $h$-th attention head,

$$\text{MAV}_h = \sum_{i \in \mathcal{M}_{av}} a_{i,h} \tag{18}$$

where $a_{i,h}$ denotes the attention value of the $h$-th attention head for the $i$-th token. We observe substantial variance across different attention heads, with the majority of heads

*Table 1.* **Performance comparison with state-of-the-art methods across video understanding benchmarks on Qwen2.5-Omni.** VideoMME is evaluated on short, medium, and long durations. Best results under each retention ratio are highlighted in bold.

| Method | Retention Ratio | FLOPs Ratio | MVBench | LongVideo Bench | MLVU | VideoMME | | | | Average | |
|---|---|---|---|---|---|---|---|---|---|---|---|
| | | | | | | Overall | Short | Medium | Long | Score | % |
| Qwen2.5-Omni-7B | 100% | 100% | 59.0 | 58.5 | 67.3 | 60.7 | 72.4 | 58.7 | 51.0 | 61.4 | 100 |
| FastV | 65% | 64% | 56.8 | 57.6 | 63.6 | 58.6 | 70.1 | 57.6 | 48.1 | 59.2 | 96.4 |
| DyCoke | 65% | 59% | 53.6 | 51.2 | 57.9 | 53.1 | 63.2 | 50.7 | 45.3 | 54.0 | 88.0 |
| PruneVID | 65% | 71% | 55.8 | 53.3 | 60.6 | 56.2 | 63.7 | 55.5 | 49.3 | 56.5 | 92.0 |
| VisionZip | 65% | 69% | 58.7 | 57.8 | 66.9 | 60.1 | 70.7 | **59.8** | 49.8 | 60.9 | 99.2 |
| FastVID | 65% | 56% | 58.6 | **58.4** | 66.2 | 60.2 | 72.0 | 58.7 | 49.8 | 60.8 | 99.0 |
| **PRIM (Ours)** | 65% | 54% | **58.8** | 58.3 | **67.1** | **60.3** | 72.2 | 58.5 | **50.2** | **61.1** | **99.5** |
| FastV | 50% | 54% | 52.2 | 54.7 | 63.3 | 57.5 | 68.9 | 56.2 | 47.3 | 56.9 | 92.7 |
| DyCoke | 50% | 44% | 53.1 | 50.3 | 57.5 | 52.6 | 61.9 | 50.6 | 45.3 | 53.4 | 87.0 |
| PruneVID | 50% | 54% | 54.1 | 52.1 | 59.2 | 54.1 | 62.9 | 52.1 | 47.2 | 54.9 | 89.4 |
| VisionZip | 50% | 51% | 56.9 | 57.5 | **66.3** | 58.5 | 70.5 | 57.1 | 47.8 | 59.8 | 97.4 |
| FastVID | 50% | 41% | 56.4 | 56.8 | 65.9 | 58.9 | 69.4 | 57.8 | **49.4** | 59.5 | 96.9 |
| **PRIM (Ours)** | 50% | 41% | **57.6** | **58.4** | 65.9 | **59.6** | 70.7 | **59.1** | 48.9 | **60.4** | **98.4** |
| FastV | 35% | 34% | 49.3 | 51.0 | 59.9 | 53.1 | 63.2 | 52.5 | 43.7 | 53.3 | 86.8 |
| DyCoke | 35% | 38% | 50.6 | 49.4 | 53.0 | 51.7 | 60.0 | 51.1 | 44.1 | 51.2 | 83.4 |
| PruneVID | 35% | 39% | 50.4 | 48.9 | 56.5 | 50.9 | 57.9 | 50.4 | 44.4 | 51.7 | 84.2 |
| VisionZip | 35% | 35% | 53.4 | 52.9 | 62.3 | 54.6 | 65.1 | 52.5 | 46.2 | 55.8 | 90.9 |
| FastVID | 35% | 28% | 54.1 | 52.3 | 61.3 | 55.1 | 66.4 | 53.3 | 45.5 | 55.7 | 90.7 |
| **PRIM (Ours)** | 35% | 28% | **54.3** | **53.2** | **62.9** | **56.2** | **66.7** | **55.1** | **46.9** | **56.7** | **92.3** |

exhibiting values close to zero, indicating limited relevance. Since heads with higher attention values are more reliable, we rank and retain the top-$k$ most salient heads by computing:

$$\mathcal{I} = \arg\max \big(\mathrm{MAV}_1, \ldots, \mathrm{MAV}_H\big)[:k] \qquad (19)$$

and derive an importance criterion

$$a_i' = \sum_{h=1}^{k} a_{i, \mathcal{I}(h)} \qquad (20)$$

to identify redundant tokens. The tokens are then sorted according to their importance scores, retaining those with higher values, while the remaining tokens are progressively pruned in subsequent layers. In practice, the MAV quantifies the degree of interaction between textual tokens and other modalities. A higher MAV indicates a greater contribution of visual and audio tokens during the generation process. Accordingly, we define the contribution as the sum of the MAV scores over the top-$k$ heads, which determines the number of tokens to be preserved:

$$K = \lambda \cdot \sum_{h=1}^{k} \sum_{i=1}^{N} a_{i, \mathcal{I}(h)} \qquad (21)$$

where $\lambda$ is a scaling constant. Consequently, the level of contribution determines the number of tokens to be retained, based on which we perform token compression for the LLM component.

## 4. Experiments

### 4.1. Experimental Settings

**Benchmarks.** We evaluate the performance of PRIM on several representative standard video understanding benchmarks, following prior protocols (Liu et al., 2025) for fair comparison. The benchmarks include MVBench (Li et al., 2024b), LongVideoBench (Wu et al., 2024), MLVU (Zhou et al., 2025), VideoMME (Fu et al., 2024), AVUT (Zhang et al., 2024a), and WorldSense (Hong et al., 2025). These benchmarks collectively cover a diverse set of scenarios, allowing for a comprehensive evaluation of PRIM's effectiveness and generalization ability.

**Comparison Methods.** To ensure consistency, we evaluated all baseline methods using the standardized multimodal evaluation framework LMMs-Eval (Zhang et al., 2025), adopting the official implementations and unified evaluation protocols provided therein. We compare the proposed PRIM with five strong baseline methods: 1) FastV (Chen et al., 2024) identifies key tokens for pruning during the pre-fill phase by predicting attention scores between token and visual tokens; 2) DyCoke (Tao et al., 2025) applies temporal merging techniques before the LLM and implements dynamic KV cache pruning during the decoding phase. We employ its first-stage TTM module to process video and audio tokens; 3) PruneVid (Huang et al., 2025) reduces video redundancy through spatiotemporal token clustering; 4) VisionZip (Yang et al., 2025) prunes tokens before the LLM by merging spatial tokens; 5) FastVID (Shen et al.,

*Table 2.* **Overall performance of various methods on the *AVUT* and *VideoMME* benchmarks.** AVUT includes IE (Information Extraction), CC (Content Counting), EL (Event Localization), CM (Character Matching), OM (Object Matching), and TM (Text Matching). Results are reported in terms of accuracy.

| Method | Retention Ratio | FLOPs Ratio | AVUT | | | | | | | VideoMME |
|---|---|---|---|---|---|---|---|---|---|---|
| | | | IE | CC | EL | CM | OM | TM | Avg. | Avg |
| *Qwen2.5-Omni-7B* | | | | | | | | | | |
| Full Tokens | 100% | 100% | 78.9 | 41.3 | 51.5 | 75.5 | 67.2 | 65.2 | 63.3 | 60.7 |
| DyCoke | 50% | 44% | 75.2 | 36.3 | 46.9 | 68.6 | 60.1 | 59.3 | 57.7 | 54.7 |
| FastVID | 50% | 42% | 77.0 | **39.7** | 47.4 | 70.0 | 62.2 | **64.9** | 60.2 | 59.4 |
| **PRIM (Ours)** | 50% | 33% | **78.6** | 39.7 | **51.7** | 74.8 | **64.5** | 64.6 | **62.3** | **59.6** |
| *Qwen2.5-Omni-3B* | | | | | | | | | | |
| Full Tokens | 100% | 100% | 78.3 | 41.3 | 46.2 | 71.4 | 66.0 | 62.7 | 61.0 | 57.0 |
| DyCoke | 35% | 29% | 72.4 | 35.5 | 39.6 | 68.8 | 60.1 | 58.9 | 55.9 | 50.6 |
| FastVID | 35% | 34% | 77.3 | **39.6** | 42.1 | 69.2 | **65.8** | 60.5 | 59.1 | 53.7 |
| **PRIM (Ours)** | 35% | 29% | **77.6** | 38.7 | **43.7** | 69.4 | 65.3 | **60.9** | **59.3** | **54.2** |

2025) partitions the video and applies density-based token pruning. We evaluate these methods using their official code under the same hardware conditions.

**Implementation Details.** We implement the proposed PRIM approach using NVIDIA A800 (80GB) GPUs. PRIM is applied to the three models, Qwen2.5-Omni (Team, 2024), LLaVA-OneVision (Li et al., 2024a), and LLaVA-Video (Zhang et al., 2024b), following the official implementations of each model. Our method is built upon the visual encoder, audio encoder, and backbone network of the large multimodal model. To ensure a fair comparison of computational costs, we use total FLOPs as the evaluation metric. Regarding parameter settings, the number of input frames is limited to 128, which are sampled from videos at a rate of 2 frames per second (2 fps). The multimodal cross-fusion module consists of four Transformer blocks. The model initially uses the experimental configuration of $\rho_{\max} = 0.7$, $\rho_{\min} = 0.3$, $\delta = 4$, $S = 0.3$, and $\rho_v = 0.6$. During the experiment, parameters will be adjusted as needed to achieve different retention rates. All experiments employ FlashAttention to reduce memory consumption. Additionally, all comparative experiments are conducted under identical experimental configurations and inference settings to ensure fair comparisons across different methods.

### 4.2. Main Results

**Results on Qwen2.5-Omni.** We compare several methods on the Qwen2.5-Omni model across multiple video understanding benchmarks, as summarized in Table 1. PRIM generally demonstrates competitive performance when compared to other strong baseline methods such as FastV, Dy-Coke, PruneVID, VisionZip, and FastVID, consistently achieving high scores across a range of retention ratios (65%, 50%, and 35%). Notably, PRIM performs strongly on medium- and long-duration videos, highlighting its ability

to handle temporal information effectively across different video lengths. PRIM's cooperative multimodal compression framework not only reduces computational cost by pruning redundant audio and visual tokens but also maintains or improves task accuracy relative to existing methods. These results demonstrate that PRIM provides an efficient and reliable solution for multimodal video understanding tasks, balancing computational efficiency with robust performance, and demonstrating the practical advantages of token compression in real-world video reasoning scenarios.

Furthermore, we compare the overall performance of different methods on the AVUT and VideoMME benchmarks, as shown in Table 2. PRIM achieves strong performance across both Qwen2.5-Omni-7B and Qwen2.5-Omni-3B under the evaluated compression settings. For Qwen2.5-Omni-7B, PRIM obtains the highest AVUT average score among compressed methods, achieving 62.3 while reducing the FLOPs ratio to 33%. It also achieves the best VideoMME average score among compressed methods, with 59.6 compared to 59.4 for FastVID and 54.7 for DyCoke. For Qwen2.5-Omni-3B, PRIM achieves the best AVUT average score among compressed methods, reaching 59.3 with a FLOPs ratio of 29%, while maintaining competitive performance on VideoMME. In particular, PRIM shows strong results on several AVUT subtasks, such as information extraction, event localization, and character matching. This improvement can be attributed to PRIM's multimodal compression framework, which reduces redundant information and optimizes performance while using comparable or lower token budgets. Its ability to maintain high accuracy across multiple tasks highlights its effectiveness in multimodal reasoning.

**Results on LLaVA-OneVision.** We evaluate the performance of various methods on the LLaVA-OV models using the WorldSense benchmark, which tests real-world multimodal understanding capabilities, as shown in Table 3. PRIM achieves the highest average accuracy among com-

*Table 3.* **Performance of different methods on LLaVA-OV model on *WorldSense* benchmark.** We conducted an evaluation of the real-world multimodal understanding capabilities of existing compression strategies on the WorldSense benchmark. The categories include Tech. (Technology & Science), Cult. (Culture & Politics), Daily. (Daily Life), Film. (Film & TV), as well as Performance, Games, Sports, and Music. Results are reported in terms of accuracy.

| Method | Retention Ratio | FLOPs(T) | Tech. | Cult. | Daily. | Film. | Performance | Games | Sports | Music | Avg |
|---|---|---|---|---|---|---|---|---|---|---|---|
| | | | | *LLaVA-OV-7B* | | | | | | | |
| Full Tokens | 100% | 74.2 | 56.7 | 51.4 | 50.6 | 55.8 | 53.3 | 47.9 | 49.8 | 45.5 | 51.4 |
| DyCoke | 50% | 37.8 | 51.9 | 49.9 | 46.8 | 51.5 | 48.7 | 44.0 | 48.8 | 40.7 | 47.8 |
| FastVID | 50% | 29.4 | 54.4 | **52.4** | **49.7** | **55.1** | 49.6 | 47.1 | 50.1 | **44.9** | 50.4 |
| **PRIM (Ours)** | 50% | 22.1 | **57.6** | 50.8 | **49.7** | 53.7 | **53.3** | **47.5** | **50.5** | 44.6 | **51.0** |
| | | | | *LLaVA-OV-0.5B* | | | | | | | |
| Full Tokens | 100% | 39.9 | 55.8 | 52.1 | 47.1 | 56.6 | 53.3 | 48.8 | 52.4 | 44.3 | 51.3 |
| DyCoke | 50% | 20.4 | 50.2 | 48.9 | 42.6 | 50.4 | 48.1 | 42.9 | 48.7 | 40.1 | 46.5 |
| FastVID | 50% | 18.1 | 53.1 | 50.2 | 43.9 | **57.6** | 49.3 | **48.8** | 49.1 | **41.3** | 49.2 |
| **PRIM (Ours)** | 50% | 15.4 | **54.4** | **51.8** | **46.0** | 56.8 | **50.0** | 47.1 | **51.9** | 41.3 | **49.9** |

*Table 4.* **Comparison of different methods on LLaVA-Video.** PRIM achieves superior performance across videos.

| Method | Retention Ratio | FLOPs (T) | VideoMME | | | |
|---|---|---|---|---|---|---|
| | | | Short | Medium | Long | Overall |
| Full Tokens | 25% | 104.7 | 76.9 | 67.4 | 53.4 | 65.9 |
| DyCoke | 25% | 28.3 | 67.8 | 59.0 | 48.9 | 58.6 |
| FastVID | 25% | 26.8 | 69.0 | 59.4 | 49.6 | 59.3 |
| **PRIM (Ours)** | 25% | 25.1 | **72.3** | **61.3** | **51.8** | **61.8** |

*Table 5.* **Inference efficiency comparison on *WorldSense*.** Our method can achieve the best model performance and the lowest memory consumption, and the greatest inference acceleration.

| Method | Memory↓ | Prefilling Time↓ | Average Accuracy↑ | Latency per Example↓ |
|---|---|---|---|---|
| | | *Qwen2.5-Omni-7B* | | |
| Full Tokens | 33G | 289ms | 51.0 | 4.56s |
| DyCoke(50%) | 32G | 278ms | 45.5 | 4.29s |
| FastVID(50%) | 28G | 184ms | 47.6 | 3.64s |
| PRIM (50%) | 27G | 109ms | 50.2 | 3.29s |
| PRIM (35%) | 25G | 91ms | 48.9 | 3.12s |
| | | *Qwen2.5-Omni-3B* | | |
| Full Tokens | 29G | 264ms | 50.5 | 3.68s |
| DyCoke(50%) | 27G | 222ms | 45.0 | 3.45s |
| FastVID(50%) | 20G | 171ms | 48.0 | 3.12s |
| PRIM (50%) | 17G | 112ms | 49.7 | 2.92s |
| PRIM (35%) | 15G | 88ms | 48.4 | 2.79s |

pressed methods on both model scales and shows strong performance across domains such as Technology, Daily Life, Performance, and Sports. Compared with DyCoke and FastVID, PRIM demonstrates a better accuracy-efficiency trade-off under the same retention ratio, highlighting its advantages for multimodal model compression. The strong performance in these categories indicates PRIM's robustness and its ability to adapt to diverse real-world scenarios.

**Results on LLaVA-Video.** We compare methods using LLaVA-Video on the VideoMME benchmark, specifically assessing performance across different video durations, as shown in Table 4. PRIM outperforms DyCoke and FastVID across all video durations and achieves the highest overall score among compressed methods. This demonstrates PRIM's superior effectiveness in preserving critical information while maintaining high accuracy across varying video lengths. These results suggest that PRIM can more effectively preserve informative tokens during compression through its cooperative compression strategies, leading to more robust performance across video lengths and making it an efficient and scalable option for video understanding tasks under computational constraints.

**Efficiency Analyses.** Table 5 assesses inference efficiency, focusing on memory usage, pre-filling time, and latency per example for the WorldSense benchmark. PRIM shows significant advantages in memory efficiency and inference speed over methods like DyCoke and FastVID. For instance,

PRIM consumes up to 12GB less memory than DyCoke and achieves nearly 30% lower inference latency. This improvement is mainly attributed to PRIM's dynamic compression strategies, which reduce the model's computational requirements while maintaining accuracy. The lower latency and memory consumption suggest PRIM's potential for real-time inference and large-scale deployment.

## 5. Conclusion

In this work, we proposed PRIM, a cooperative compression and inference framework for efficient audio-visual reasoning in large multimodal models. Motivated by an attention-based analysis of modality imbalance and layer-wise redundancy, PRIM systematically compresses multimodal tokens by combining early text-conditioned audio-visual fusion with signal-aware compression and instruction-aware adaptation strategies. By reducing redundant audio and video tokens while preserving semantically critical information, PRIM significantly improves inference efficiency. Extensive experiments across multiple audio-visual benchmarks demonstrate that PRIM consistently achieves stable and superior efficiency–accuracy trade-offs across diverse tasks.

## Impact Statement

This paper presents work whose goal is to increase understanding of deep learning, which may lead to advancements in the field of Machine Learning. There are many potential societal consequences of our work, none of which we feel must be specifically highlighted here.

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

## A. Processing Expense

In the experimental section, we evaluate the floating-point operations (FLOPs), defined as the total computational cost introduced by audio and video tokens during both the prefill and decoding stages. In our analysis, we consider a Transformer layer architecture composed of multi-head attention (MHA) and feed-forward network (FFN) modules. Here, $n$ denotes the number of tokens, $d$ represents the hidden state dimension, and $m$ is the intermediate dimension of the FFN. During the prefill stage, the total FLOPs can be approximately expressed as $4nd^2 + 2n^2d + 2ndm$. For the decoding stage, taking into account the significant contribution of the key-value (KV) cache, the computational cost over $R$ iterations (i.e., predicting $R$ tokens) is given by $R(4d^2 + 2dm) + 2\sum_{i=1}^{R}(n+i)d$. In our experiments, we uniformly set $R = 100$. Therefore, for an LLM with $T$ Transformer layers, the total FLOPs can be formulated as:

$$\text{FLOPs} = T\left(4nd^2 + 2n^2d + 2ndm\right) + TR\left(\left(4d^2 + 2dm\right) + 2\left(dn + \frac{d(R+1)}{2}\right)\right) \tag{22}$$

## B. Ablation Study

Unless otherwise noted, all ablation experiments are conducted on Qwen2.5-Omni and evaluated on the video understanding benchmarks in the Experiments section.

**Impact of Hyperparameters $S$ and $\rho_v$ on Performance.** The sensitivity of the proposed method to the hyperparameters $S$ and $\rho_v$ is investigated in Figure 5. The experiments are conducted on the VideoMME benchmark to analyze how different levels of modality-specific token compression affect the overall model performance. In the left subfigure, $\rho_v$ is fixed while the retention ratio of audio tokens $S$ is varied. The results indicate that an appropriate audio token retention rate leads to optimal performance, whereas overly aggressive audio compression consistently degrades task accuracy. This observation suggests that, although audio tokens exhibit a certain degree of redundancy, they play a critical role in preserving fine-grained temporal features and semantic information. Excessive removal of audio tokens disrupts cross-modal alignment, thereby weakening the model's downstream reasoning capability. In the right subfigure, $S$ is fixed and the video pruning ratio $\rho_v$ is adjusted. A similar trend is observed: when the video pruning ratio exceeds a certain threshold, model performance drops significantly. However, compared to the audio modality, the performance curve with respect to $\rho_v$ is noticeably smoother, indicating that visual tokens are more redundant and can tolerate more aggressive pruning without causing abrupt performance degradation.

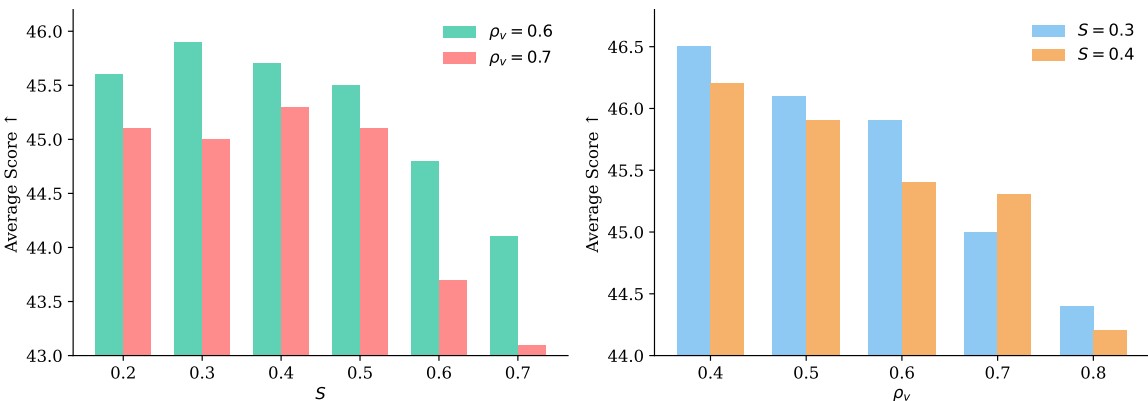

*Figure 5.* **Impact of hyperparameters $S$ and $\rho_v$ on performance.** All experiments illustrated in the figure are conducted on the VideoMME benchmark. We analyze the impact of varying parameters $S$ and $\rho_v$ on model performance, respectively. It can be observed that excessive pruning of either modality will exert a negative impact on model performance. However, a proper balance between audio and video token pruning yields the optimal performance.

**Performance Impact of $\delta$ and Fusion Layers.** The impact of the token merging parameter $\delta$ and the number of fusion layers on both performance and efficiency is analyzed in Figure 6. The left subplot evaluates model accuracy on AVUT and WorldSense under different values of $\delta$, which controls the number of non-significant tokens merged into each representative node. We observe that moderate values of $\delta$ consistently yield the best performance across both benchmarks. When $\delta$ is too small, the representation remains overly fragmented, limiting the model's ability to aggregate semantically related information. Conversely, overly large $\delta$ leads to excessive merging, which blurs fine-grained distinctions and results in

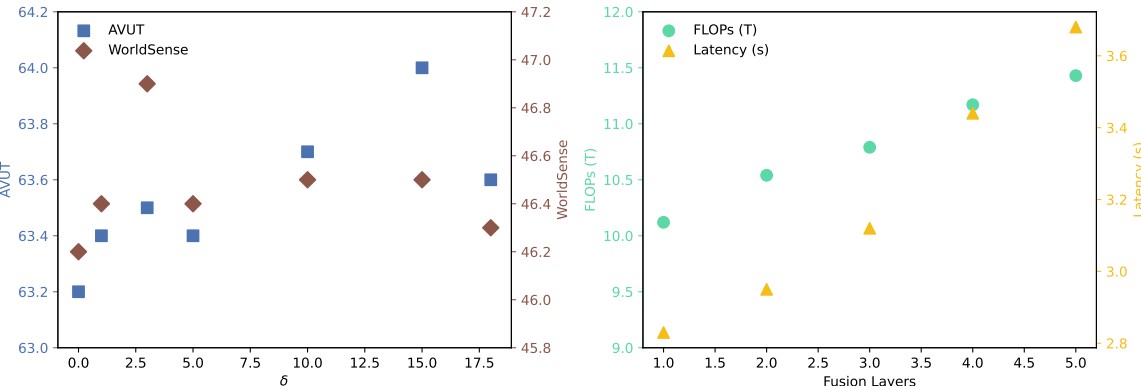

*Figure 6.* **Performance impact of $\delta$ and fusion layers.** $\delta$ denotes the number of tokens merged into each representative token. The right subplot reports FLOPs and latency under different numbers of fusion layers.

*Table 6.* **Ablation study of PRIM components.** To validate the effectiveness of the proposed method, we evaluate the impact of individual compression components on the final model accuracy, including Multimodal Cross-Fusion (MCF), Attention-Guided Selection (AGS), and Frequency-Aware Compaction (FAC). Results are reported in terms of accuracy.

| Settings | | | AVUT | VideoMME | WorldSense |
|---|---|---|---|---|---|
| MCF | AGS | FAC | Avg | Avg | Avg |
| × | × | × | 63.3 | 60.7 | 51.0 |
| ✓ | × | × | 61.6 | 59.1 | 49.7 |
| × | ✓ | ✓ | 62.1 | 59.4 | 50.1 |
| ✓ | ✓ | ✓ | 63.0 | 60.1 | 50.7 |

noticeable performance degradation. This trend indicates that an appropriate level of structural abstraction is critical for preserving informative multimodal cues while suppressing redundancy. The right subplot examines the effect of fusion depth on computational cost. Increasing the number of fusion layers leads to a near-linear growth in FLOPs and inference latency, reflecting the additional cross-modal interactions introduced by deeper fusion. While deeper fusion slightly enhances representation capacity, the marginal performance gains diminish beyond a small number of layers, whereas computational overhead continues to accumulate. This observation suggests that early-stage fusion is sufficient to capture the majority of cross-modal dependencies, and deeper fusion mainly introduces redundant computation.

**Ablation Study of PRIM Components.** Table 6 presents an ablation study examining the contribution of each compression component in PRIM, including Multimodal Cross-Fusion (MCF), Attention-Guided Selection (AGS), and Frequency-Aware Compaction (FAC). The first row corresponds to the uncompressed setting, which retains all tokens and thus serves as an upper-bound reference in terms of accuracy, but incurs substantially higher memory and computational costs. Under the compressed setting, enabling all components achieves the best performance across all benchmarks, demonstrating the effectiveness of the proposed compression framework. Removing AGS and FAC while retaining only MCF leads to a noticeable performance drop, suggesting that cross-modal fusion alone is insufficient to effectively suppress redundant tokens without explicit importance modeling. Similarly, removing MCF while keeping AGS and FAC also leads to performance degradation, highlighting the importance of explicit cross-modal interaction for maintaining semantic information during compression. Overall, these results show that the three components are complementary and jointly help the model achieve a favorable trade-off between efficiency and accuracy.

**Comparison of Predefined and Instance-Adaptive Pruning Ratios.** We compare predefined pruning ratios with the proposed instance-adaptive strategy under the same training-free setting in Table 7. We also report an ablation of attention-head filtering for the predefined pruning baseline. The results are evaluated on AVUT, VideoMME, and WorldSense, together with the average token budgets, retained token counts, and pruning layer configurations. For predefined pruning ratios, the performance is noticeably affected by the pruning configuration. Although fixed retained-token budgets can achieve reasonable results, their effectiveness varies across benchmarks, suggesting limited flexibility in handling heterogeneous input distributions. Introducing attention-head filtering improves the predefined baseline, but the performance remains constrained

*Table 7.* Comparison of predefined pruning ratio and instance-adaptive ratio with or without attention-head filtering under the same training-free setting. Pruning layer indexes specify the LLM layers at which tokens are pruned. Token counts marked with (*) are instance-adaptive averages.

| Pruning Strategy | Attn-head Filtering | Avg. Tokens | Pruning Layer Indexes | Avg. Retained Tokens | AVUT | VideoMME | WorldSense |
|---|---|---|---|---|---|---|---|
| Upper Bound | - | 1152 | - | - | 63.3 | 60.7 | 51.0 |
| Predefined Ratio | × | 576 | [1] | [560] | 60.4 | 56.2 | 50.2 |
|  | √ | 576 | [1] | [560] | 60.9 | 57.2 | 50.3 |
|  | √ | 480 | [4, 12, 20] | [780, 345, 155] | 61.0 | 59.1 | 50.3 |
| Our Adaptive Ratio | √ | 620* | [4, 12, 20] | [1040*, 510*, 245*] | 62.8 | 60.5 | 50.8 |
|  | √ | 500* | [4, 12, 20] | [820*, 360*, 165*] | 62.2 | 59.9 | 50.5 |
|  | √ | 400* | [4, 12, 20] | [620*, 240*, 110*] | 61.7 | 59.3 | 50.4 |

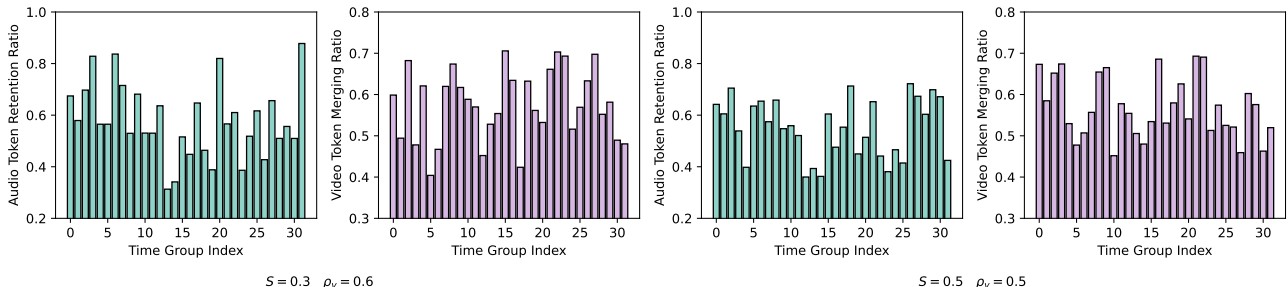

*Figure 7.* **Visualization of dynamic pruning ratios.** This figure illustrates how the audio token retention ratio governs the allocation of video token compression and pruning. Specifically, for temporal windows with higher audio retention ratios, lower video pruning ratios are assigned, and vice versa.

by the fixed pruning schedule. In contrast, the instance-adaptive pruning strategy consistently outperforms the predefined baselines across all three benchmarks. Notably, even with a lower average token budget, e.g., 400 average tokens compared with the 480-token predefined setting, the adaptive strategy still achieves better performance on AVUT, VideoMME, and WorldSense. This indicates that dynamically allocating token budgets according to instance-level characteristics enables more effective utilization of retained tokens. Moreover, the adaptive strategy maintains competitive performance under different average token budgets, demonstrating improved robustness to pruning intensity.

Furthermore, we present additional visualizations for the dynamic pruning ratio allocation, as shown in Figure 7.

We visualized the spectrogram heatmap as shown in Figure 8 to illustrate the significant concentration of energy in the low-frequency components of the visual features.

# C. Scalability of PRIM

PRIM is motivated by our systematic observation of how multimodal audiovisual tokens are organized in existing large multimodal models, particularly the widely adopted time-slice-based sequence construction paradigm. In practice, most current models follow a similar design: continuous audio and video streams are first segmented into a series of short temporal intervals, within which tokens from different modalities are fused or concatenated, and the resulting unified sequence is then fed into a large language model for processing. Owing to the high degree of architectural consistency of this paradigm across existing systems, PRIM can be naturally transferred and applied to a broad range of mainstream model configurations. At the same time, we acknowledge that LMMs are still in a rapid early stage of development, which raises a potential concern: if future models no longer explicitly rely on time-window-based token concatenation, will PRIM remain applicable? Our answer is affirmative. PRIM does not depend on any specific implementation detail, but instead builds upon the intrinsic temporal locality of audiovisual data. Within any sufficiently short temporal span, audio and visual signals are typically highly synchronized and semantically coherent, while also containing substantial redundant information that can be effectively compressed. Therefore, regardless of how model architectures evolve, the core principle

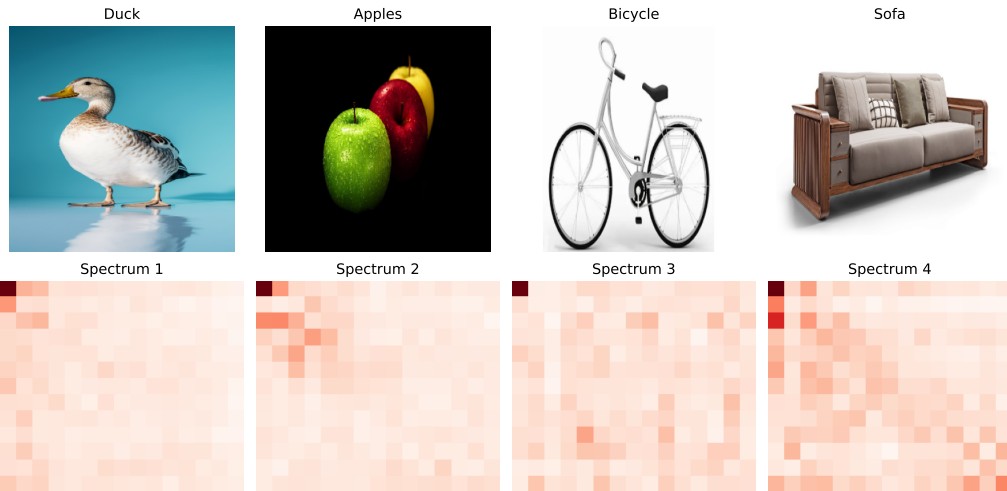

*Figure 8.* **Visualization of the frequency spectrum heatmap computed from the output of the visual encoder.** It can be seen that visual features exhibit significant energy concentration in the low-frequency components.

of PRIM—leveraging the correlations among multimodal tokens within local temporal regions to guide effective token selection and reduction—remains fundamentally valid and continues to hold practical value.

In addition, PRIM decouples its compression and fusion modules from the backbone network, ensuring that the original architecture and parameters of the LLM remain entirely untouched. This design allows PRIM to be directly compatible with existing inference acceleration frameworks, while also facilitating independent optimization or replacement of audio and visual processing modules in future extensions, thereby enhancing the overall extensibility of the system. Through this externalized processing design, PRIM significantly improves adaptability across diverse hardware and deployment environments, while preserving the effectiveness of multimodal interactions.

## D. Challenges of Multimodal Token Compression

Throughout the evolution of multimodal models, controlling the scale of tokens has emerged as a critical factor affecting inference efficiency. Prior work in the visual domain has demonstrated that aggressively reducing the number of tokens is feasible under single-modality settings; however, such experience does not readily transfer to large multimodal models. The fundamental reason lies in the fact that audiovisual information plays inherently unequal semantic roles across different application scenarios, and the model's reliance on audio and visual modalities is highly task-dependent. Moreover, audio signals are typically represented as high-dimensional features with strong temporal dependencies and lack explicit spatial structure, making them more susceptible to critical information loss during compression. In addition, current mainstream architectures have already been highly optimized with respect to token utilization, rendering further reductions based solely on conventional pruning strategies unlikely to yield substantial gains. This indicates that efficient multimodal token reduction is not merely an engineering challenge, but a structural one. Nevertheless, accurate modeling of complex video content intrinsically requires the joint participation of both audio and visual information during inference, which makes the design of a compression mechanism that balances efficiency and representational capacity particularly urgent. Against this backdrop, PRIM introduces a unified compression approach for audiovisual tokens. We hope that this method can offer a new perspective on multimodal token processing and provide a reference solution for future research.

## E. Related Work

**Large Multimodal Models.** Large multimodal models extend large language models to jointly process visual, auditory, and textual inputs, enabling unified reasoning across multiple modalities. Recent representative models, such as LLaVA-Video, Qwen-VL, and InternVL, have demonstrated strong performance on a wide range of audio-visual understanding tasks, including event localization, temporal reasoning, and cross-modal grounding (Zhang et al., 2024b; Bai et al., 2023; Wang et al., 2025b). Beyond these, newer approaches like SlowFast-LLaVA-1.5 and QMAVIS have been proposed to enhance

efficiency and long-form video-audio understanding, showing competitive capabilities on extended video benchmarks (Xu et al., 2025; Lin et al., 2026). To assess real-world performance more rigorously, recent benchmarks such as ALLVB and InfiniBench provide large-scale long video evaluation frameworks with diverse tasks and challenging temporal contexts (Tan et al., 2025; Ataallah et al., 2025). Despite these advances, existing LMMs still face efficiency bottlenecks when processing long-form audio-visual inputs, as the number of multimodal tokens can easily scale to tens of thousands, motivating interest in improving inference efficiency without sacrificing reasoning capability in real-world deployment scenarios.

**Token Compression and Pruning.** Token compression and pruning have been extensively studied in both vision and language domains as a means to reduce computational cost. In vision transformers, numerous works propose token pruning or merging strategies based on attention scores, spatial redundancy, or similarity-based criteria, achieving significant acceleration while preserving accuracy (Zeng et al., 2025; Li et al., 2025c; Jeddi et al., 2025; Götz et al., 2025; Su et al., 2026). Similarly, in language models, token pruning and early exiting techniques have been explored to reduce inference latency by removing low-importance tokens or skipping layers (Li et al., 2025a; Wang et al., 2025a; Li et al., 2025b; Wen et al., 2025). However, most existing approaches are designed for single-modality settings or apply uniform pruning rules across tokens. When directly extended to multimodal models, such strategies often ignore modality-specific characteristics and can lead to suboptimal compression or information loss.

In contrast, PRIM differs from existing methods by introducing an efficient cooperative compression framework that integrates early text-conditioned audio-visual fusion, attention-guided and frequency-aware token reduction, and instruction-aware adaptation inside the language model. This design enables principled and flexible compression guided by multimodal attention and task relevance, resulting in superior efficiency–accuracy trade-offs across a wide range of multimodal benchmarks.

# F. Limitations

Although this work is the first to substantially improve the inference efficiency of large multimodal models through multi-perspective collaborative compression, several open issues remain for future investigation. First, the reliance on audio and visual information varies significantly across application scenarios. How to adaptively allocate compression ratios between the two modalities in a general and robust manner across diverse tasks remains an open problem. Second, the current approach is primarily designed for offline inference and does not directly support online or streaming audiovisual inputs with uncertain durations. Developing inference mechanisms that can effectively exploit audio information during streaming video understanding constitutes an important direction for future research. Third, the parameter scale of ultra-large models continues to impose substantial computational and deployment costs. Therefore, jointly designing audiovisual token compression with other model efficiency techniques holds promise for further unlocking the practical potential of large multimodal models in real-world applications.

