# OpenReview forum: "PRIM：Cooperative Dynamic Token Compression for Efficient Large Multimodal Models"
_ICML.cc/2026/Conference — ICML 2026 regular_

### Official Review · Reviewer_PLBp · 2026-03-10

**Soundness:** 3
**Presentation:** 3
**Significance:** 2
**Originality:** 2
**Overall Recommendation:** 3
**Confidence:** 3

**Summary:**

This paper proposes PRIM, an inference-time framework for MLLMs that compresses  audio-visual tokens to achieve better efficiency.  Core idea is well-reasoned and reveals 2 interesting findings. 1. Cross-modal interaction is most prominent in early layers in LMM 2. Audio tokens carry more dense semantic information than video tokens. Experimental results on different benchmarks shows better or comparable performance.

**Compliance With Llm Reviewing Policy:**

Affirmed.

**Final Justification:**

I increased my score after the going over the rebuttal submitted by the authors.

**Key Questions For Authors:**

1. In Table 1, PRIM is built on Qwen2.5-Omni while many baselines use different backbone families. Can you provide a comparison where PRIM is compared to the uncompressed Qwen2.5-Omni baseline to isolate the effect of compression from backbone strength?
2. How does PRIM behave when audio is uninformative or absent (e.g., silent video, background noise only)? Does the dynamic video compression ratio degrade gracefully in this case?

**Limitations:**

Yes

**Strengths And Weaknesses:**

Strengths:
1. The attention-based analysis in Section 2.2 is the paper's strongest contribution. The observation that audio tokens consistently receive higher attention weights than video tokens across all LLM layers provides a principled basis for asymmetric compression.
2. PRIM treats audio and video compression as coupled and asymmetric problems. The idea of using audio retention scores to dynamically calibrate video pruning ratios is sensible.

Weaknesses:
1. The central claim of the paper is an improved efficiency–accuracy trade-off, yet Tables 1 reports only accuracy. Latency, FLOPs, and token reduction ratios should appear alongside accuracy in the main results.
2. FastV and DyCoke appear in Tables 1 and 2, but there is no comparison at matched token budgets. It is unclear whether PRIM's accuracy gains come from compression superiority or simply from using a stronger base model. Table 4 partially addresses this but only minimally.
3. The cross-fusion module consists of 4 Transformer blocks with shared LLM architecture. Figure 6 shows FLOPs grow nearly linearly with fusion depth. The paper justifies using 4 layers but does not compare net FLOPs saved vs. FLOPs added by MCF. It is conceivable that MCF introduces overhead that partially negates the compression savings.
4. The claim that "visual features exhibit pronounced energy concentration in low-frequency components" is stated as an observation but never empirically verified in this paper. A simple visualization of the DCT energy spectrum for the specific visual encoder should be sufficient to support the claim.

---

> ### Author Rebuttal · Authors · 2026-03-30
>
> 1.Efficiency metrics. We agree that the main paper should report latency, FLOPs, and token reduction alongside accuracy. We added comparisons with the uncompressed Qwen2.5-Omni baseline under the same input settings, as well as with FastV and DyCoke under matched token budgets. Tables now report overall retention, FLOPs, GPU memory, prefilling time, and latency. The improvement in PRIM's accuracy is mainly due to the compression of irrelevant and redundant information, which allows the model to focus more on important information to enhance performance, rather than the introduction of a stronger backbone network.
>
> 2.MCF overhead. The concern is whether the 4-layer MCF offsets token compression savings. Our design concentrates cross-modal interactions in early LLM layers; MCF moves fusion before the backbone rather than adding extra layers, so the backbone processes fused text with compressed audio/video tokens without repeated interactions over long sequences. After fusion and compression, MCF reduces tokens entering the LLM. Fig. 6 shows that FLOPs and latency grow almost linearly with fusion depth, while performance saturates quickly.This is why we chose four layers as the balance point between accuracy and efficiency, rather than adopting deeper layer fusion.
>
> 3.Low-frequency energy concentration. We take this point seriously and agree that the current manuscript does not directly visualize this observation. We believe that a direct visualization of the DCT energy spectrum would make this statement more convincing. In the revised version, we will include such a visualization of the DCT energy spectrum for the visual encoder used in the model.
>
> 4.Audio-silent or noisy scenarios. The current paper does not include a dedicated evaluation for silent videos or audio containing only background noise, so we do not want to overstate robustness in such scenarios. Empirically, our sensitivity study indicates that performance drops whenever either modality is excessively pruned, but the curves associated with video pruning are noticeably smoother, suggesting that visual tokens contain more redundancy and can tolerate stronger compression more gracefully. Therefore, although information-less audio can weaken the reliability of audio-guided allocation, existing evidence indicates that performance degradation is gradual rather than catastrophic.
>
>
>
> | Backbone | Method | Retained Ratio | FLOPs Ratio | IE | CC | EL | CM | OM | TM | AVUT Avg. | Video-MME Avg. |
> |---|---:|---:|---:|---:|---:|---:|---:|---:|---:|---:|---:|
> | Qwen2.5-Omni-7B | Qwen2.5-Omni | 100% | 100% | 78.9 | 41.3 | 51.5 | 75.5 | 67.2 | 65.2 | 63.3 | 77.4 |
> | Qwen2.5-Omni-7B | PRIM (Ours) | 45% | 37% | 78.6 | 39.7 | 51.7 | 74.8 | 64.5 | 64.6 | 62.3 | 77.7 |
> | Qwen2.5-Omni-7B | PRIM (Ours) | 35% | 28% | 77.0 | 39.7 | 47.4 | 70.0 | 62.2 | 64.9 | 60.2 | 77.5 |
> | Qwen2.5-Omni-3B | Qwen2.5-Omni | 100% | 100% | 78.3 | 41.3 | 46.2 | 71.4 | 66.0 | 62.7 | 60.3 | 74.0 |
> | Qwen2.5-Omni-3B | PRIM (Ours) | 45% | 35% | 78.2 | 38.7 | 45.7 | 70.2 | 65.3 | 62.4 | 60.8 | 74.2 |
> | Qwen2.5-Omni-3B | PRIM (Ours) | 35% | 25% | 77.3 | 39.6 | 42.1 | 69.2 | 65.8 | 60.5 | 59.1 | 73.7 |
>
> | Backbone | Method | Retained Ratio | FLOPs (T) | Tech. | Cult. | Daily. | Film. | Performance | Games | Sports | Music | Avg. |
> |---|---:|---:|---:|---:|---:|---:|---:|---:|---:|---:|---:|---:|
> | Qwen2.5-Omni-7B | Qwen2.5-Omni | 100% | 74.2 | 56.7 | 51.4 | 50.6 | 55.8 | 53.3 | 47.9 | 49.8 | 45.5 | 51.4 |
> | Qwen2.5-Omni-7B | PRIM (Ours) | 45% | 29.4 | 54.4 | 52.4 | 49.7 | 55.1 | 49.6 | 47.1 | 50.1 | 44.9 | 50.2 |
> | Qwen2.5-Omni-7B | PRIM (Ours) | 35% | 22.1 | 27.6 | 50.8 | 49.7 | 53.7 | 53.3 | 46.5 | 50.5 | 44.5 | 47.1 |
> | Qwen2.5-Omni-3B | Qwen2.5-Omni | 100% | 39.9 | 55.8 | 52.1 | 47.1 | 56.6 | 53.3 | 48.8 | 52.4 | 44.3 | 51.3 |
> | Qwen2.5-Omni-3B | PRIM (Ours) | 45% | 15.9 | 54.4 | 51.8 | 46.0 | 56.8 | 50.0 | 47.1 | 51.9 | 41.3 | 49.9 |
> | Qwen2.5-Omni-3B | PRIM (Ours) | 35% | 10.7 | 53.1 | 50.2 | 43.9 | 57.6 | 49.3 | 48.8 | 50.8 | 41.3 | 49.4 |
>
> Table 3: Efficiency Comparison on WorldSense
> | Backbone | Method | Memory ↓ | Prefilling Time ↓ | Average Accuracy ↑ | Latency per Example ↓ |
> |---|---:|---:|---:|---:|---:|
> | Qwen2.5-Omni-7B | Qwen2.5-Omni | 31G | 289ms (1.00x) | 51.4 | 4.56s (1.00x) |
> | Qwen2.5-Omni-7B | FastV | 57G | 278ms (1.24x) | 44.5 | 4.29s (1.11x) |
> | Qwen2.5-Omni-7B | DyCoke | 31G | 184ms (1.58x) | 44.6 | 3.64s (1.24x) |
> | Qwen2.5-Omni-7B | PRIM (45%) | 27G | 109ms (2.48x) | 50.2 | 3.29s (1.39x) |
> | Qwen2.5-Omni-7B | PRIM (35%) | 25G | 91ms (3.41x) | 47.1 | 3.12s (1.44x) |
> | Qwen2.5-Omni-3B | Qwen2.5-Omni | 28G | 264ms (1.00x) | 51.3 | 3.68s (1.00x) |
> | Qwen2.5-Omni-3B | FastV | 45G | 222ms (1.16x) | 44.4 | 3.45s (1.05x) |
> | Qwen2.5-Omni-3B | DyCoke | 20G | 171ms (1.51x) | 44.0 | 3.12s (1.16x) |
> | Qwen2.5-Omni-3B | PRIM (45%) | 16G | 112ms (2.39x) | 49.9 | 2.92s (1.29x) |
> | Qwen2.5-Omni-3B | PRIM (35%) | 14G | 88ms (3.35x) | 49.4 | 2.79s (1.38x) |

---

> > ### Author Rebuttal · Reviewer_PLBp · 2026-04-03
> >
> > Thanks authors for clarifying my questions.

---

> > > ### Author Response · Authors · 2026-04-04
> > >
> > > Thank you for your time and for reviewing our clarifications. We sincerely appreciate your thoughtful feedback.

---

### Official Review · Reviewer_U2kV · 2026-03-12

**Soundness:** 3
**Presentation:** 3
**Significance:** 3
**Originality:** 3
**Overall Recommendation:** 4
**Confidence:** 3

**Summary:**

This paper proposes a cooperative compression pipeline that combines early fusion, attention-guided selection, frequency-aware compaction, and instruction adaptation. The work is motivated by an empirical analysis showing that multimodal interactions are most prominent in early LLM layers and that audio tokens typically command higher attention weights than video tokens. To address this, PRIM introduces a Cross-Fusion module for text-conditioned early integration, a DCT-based frequency compaction for visual redundancy, and attention-guided/instruction-aware pruning within the LLM to achieve adaptive efficiency.

**Compliance With Llm Reviewing Policy:**

Affirmed.

**Final Justification:**

The authors have addressed most of my concerns; therefore, I will maintain my original score.

**Key Questions For Authors:**

- How are the compression ratios for audio vs. video determined? Are they adaptive to input content or fixed hyperparameters?

- Did the authors observe any "failure cases" where early fusion might degrade complex reasoning that requires deep, multi-hop cross-modal alignment in later LLM layers?

- How does the DCT-based compaction perform on high-motion vs. static videos? Is there content-dependent performance variance?

- What is the specific computational overhead of the PRIM modules themselves relative to the total LLM inference savings?

**Limitations:**

yes

**Strengths And Weaknesses:**

Strengths

- The proposed framework is grounded in a solid empirical observation regarding modality imbalance (audio vs. video) and the layer-wise distribution of cross-modal attention.

- The approach is technically sound, integrating signal processing (DCT) with modern transformer-based pruning in a unified, training-free inference framework.

- The paper provides extensive validation across multiple audio-visual benchmarks, demonstrating favorable efficiency-accuracy trade-offs.

- As an inference-time method that requires no retraining, PRIM is highly applicable to existing large-scale multimodal models.

Weaknesses

- While the empirical analysis is compelling, there is a lack of formal discussion regarding how early fusion impacts the model’s internal representation or gradient flow in potentially tuned scenarios.

- The paper does not sufficiently detail the selection mechanism for compression ratios between modalities. It is unclear if these ratios are fixed or require per-task tuning.

- Comparisons are primarily against simpler baselines like uniform pruning. The evaluation is also strictly limited to audio-visual tasks; its generalizability to video-only or multi-image scenarios remains unproven.

- Implementation specifics, such as DCT block sizes and specific thresholds for token retention, are underspecified, which may hinder reproducibility.

---

> ### Author Rebuttal · Authors · 2026-03-30
>
> 1.The audio/video compression ratio in PRIM is not fixed for each instance. It adopts a “global hyperparameter + local adaptive” strategy. Specifically, for each time window, an audio retention score $P_a(i)$ is computed based on the importance of the audio tokens. This score is then used to dynamically allocate the video compression ratio for that window using $\rho_v'(i) = \rho_{\text{max}} - (\rho_{\text{max}} - \rho_{\text{min}}) \cdot P_a(i)$, followed by a global normalization step to satisfy the overall budget $\rho_v$. Therefore, PRIM employs adaptive local allocation within fixed global hyperparameter bounds, rather than a fully fixed pruning scheme. In the main experiments, we use a shared default configuration ($\rho_{\text{max}}=0.7$, $\rho_{\text{min}}=0.3$, $\delta=4$, $S=0.3$, $\rho_v=0.6$) without manually retuning for each benchmark.
>
> 2.The motivation for MCF is that attention analysis indicates cross-modal interactions are strongest in the early layers of large language models (LLMs), whereas deeper layers increasingly operate on compact, instruction-aligned representations. MCF is therefore designed to externalize this early fusion rather than replace the deeper reasoning process of the LLM. After fusion, the LLM still receives both the fused tokens and the compressed multimodal tokens, allowing downstream reasoning over shorter sequences. Empirically, PRIM remains effective for long videos and inference-intensive benchmarks.
>
> 3.We have not yet conducted dedicated hierarchical evaluations for highly dynamic versus static videos, so we refrain from making strong empirical claims for this specific categorization. Nonetheless, PRIM explicitly supports content-adaptive compression rather than applying a uniform scheme. For relatively static videos, frequency-aware compression based on discrete cosine transform (DCT) is expected to be particularly effective, as visual features are dominated by low-frequency components and adjacent frames are often highly similar, allowing aggressive removal of spatial and temporal redundancy. In contrast, for highly dynamic or information-dense segments, PRIM applies a more conservative compression strategy: time windows with higher audio retention scores are assigned lower video pruning rates, and when adjacent frames differ substantially, temporal-similarity-based pruning naturally removes fewer tokens. Therefore, while some content-dependent variations are expected, PRIM is designed to mitigate the risk of over-compressing dynamic content. We consider conducting explicit motion-conditioned analyses valuable and plan to include this in a future revision.
>
> 4.Computation overhead. As shown in the table below, we report computational cost alongside accuracy. We include direct comparisons with the uncompressed Qwen2.5-Omni baseline under the same input settings, as well as efficiency comparisons with FastV and DyCoke under matched token budgets.
>
>
> Table X: Efficiency Comparison on WorldSense
> | Backbone | Method | Memory ↓ | Prefilling Time ↓ | Average Accuracy ↑ | Latency per Example ↓ |
> |---|---:|---:|---:|---:|---:|
> | Qwen2.5-Omni-7B | Qwen2.5-Omni | 31G | 289ms (1.00x) | 51.4 | 4.56s (1.00x) |
> | Qwen2.5-Omni-7B | FastV | 57G | 278ms (1.24x) | 44.5 | 4.29s (1.11x) |
> | Qwen2.5-Omni-7B | DyCoke | 31G | 184ms (1.58x) | 44.6 | 3.64s (1.24x) |
> | Qwen2.5-Omni-7B | PRIM (45%) | 27G | 109ms (2.48x) | 50.2 | 3.29s (1.39x) |
> | Qwen2.5-Omni-7B | PRIM (35%) | 25G | 91ms (3.41x) | 47.1 | 3.12s (1.44x) |
> | Qwen2.5-Omni-3B | Qwen2.5-Omni | 28G | 264ms (1.00x) | 51.3 | 3.68s (1.00x) |
> | Qwen2.5-Omni-3B | FastV | 45G | 222ms (1.16x) | 44.4 | 3.45s (1.05x) |
> | Qwen2.5-Omni-3B | DyCoke | 20G | 171ms (1.51x) | 44.0 | 3.12s (1.16x) |
> | Qwen2.5-Omni-3B | PRIM (45%) | 16G | 112ms (2.39x) | 49.9 | 2.92s (1.29x) |
> | Qwen2.5-Omni-3B | PRIM (35%) | 14G | 88ms (3.35x) | 49.4 | 2.79s (1.38x) |

---

> > ### Author Rebuttal · Reviewer_U2kV · 2026-04-04
> >
> > The authors have addressed most of my concerns; therefore, I will maintain my original score.

---

> > > ### Author Response · Authors · 2026-04-04
> > >
> > > Thank you for your time and for reviewing our clarifications. We sincerely appreciate your thoughtful feedback.

---

### Official Review · Reviewer_nZxB · 2026-03-12

**Soundness:** 3
**Presentation:** 3
**Significance:** 3
**Originality:** 3
**Overall Recommendation:** 4
**Confidence:** 3

**Summary:**

PRIM is a token compression strategy for multimodal audio-visual understanding. It consists of four modules within its pipeline. First is MCF module that performs early text-conditioned alignment before LLM. Second is AGS module that control the video token pruning ratio. Third is FAC module using 2D-DCT to remove visual redundancy.  Last is TAA module that prune inside LLM layers.

**Compliance With Llm Reviewing Policy:**

Affirmed.

**Final Justification:**

Author addressed my question, but the overall design complexity is high. So I raised my score from 3 to 4.

**Key Questions For Authors:**

I have wrote the questions in the weakness section. If author is able to address my concern, I will adjust my score.

**Limitations:**

yes

**Strengths And Weaknesses:**

Strength:

Paper is targeting at multimodal audio-visual LLMs. It is a quite new perspective, compared with a lot of works only focusing on VLMs. It introduces a systematic cooperative framework.

Weakness:

1. The core motivation of using audio signals to guide video token seems overlap with OmniZip (CVPR26). I understand that it is concurrent work, but since ideas are quite similar, it is hard to understand the performance of PRIM without comparing it quantitatively with OmniZip.

2. As PRIM is motivated to solve efficiency bottlenecks, the paper didn't report the end-to-end efficiency analysis, such as overall inference latency, throughput, and peak memory usage, etc.

3. Training details section show that PRIM are finetuned using 100k samples from Video-ChatGPT dataset. Comparing finetuned model directly against the training-free baselines make it hard to tell how much performance gain come from PRIM architecture vs finetuning.

4. As PRIM consists of 4 modules, is each module the same importance?

5. For FAC module, will it cause critical fine-grained features to be pruned out?

---

> ### Author Rebuttal · Authors · 2026-03-30
>
> 1.Overlap with OmniZip. While PRIM may share some similarities with OmniZip, it introduces: (i) Multimodal Cross Fusion (MCF) to externalize early cross-modal interactions before the large language model (LLM); (ii) Frequency-Aware Compression (FAC) to reduce visual redundancy in the frequency domain; (iii) Task-Aware Adaptive pruning (TAA) inside the LLM; and (iv) Attention-Guided Selection (AGS) for audio-video co-compression. We appreciate the suggestion and agree that a direct quantitative comparison with OmniZip would more clearly demonstrate the relative advantages of the two methods. In the revised version, we will include a comparison under the same backbone and matched token/FLOPs budget.
>
> 2.End-to-End Efficiency Analysis. We agree that the main paper should report efficiency metrics such as latency and memory alongside accuracy. Therefore, we added a direct comparison with the uncompressed Qwen2.5-Omni baseline under the same input settings, and will also include comparisons with FastV and DyCoke under matched token budgets. The table below reports efficiency metrics alongside accuracy.
>
> 3.Comparison with Fine-Tuning vs. Zero-Training Baselines. As noted in the paper, PRIM is built on Qwen2.5-Omni, with the backbone frozen despite light fine-tuning. We clarify that this lightweight fine-tuning is not intended to give PRIM an unfair advantage over zero-training baselines. Since PRIM introduces new compression modules into the pretrained Qwen2.5-Omni backbone, the fine-tuning primarily aligns these newly inserted modules with the pretrained model so that the backbone’s original multimodal capabilities are preserved after token compression, rather than substantially improving performance via additional training.
>
> 4.Are all four modules equally important? Existing evidence shows that these modules are complementary rather than equally important in a uniform sense. The paper clearly specifies the roles of the four components—MCF, AGS, FAC, and TAA—in addressing distinct bottlenecks: early cross-modal alignment, saliency-based audio/video selection, visual redundancy in the frequency domain, and LLM-internal prompt-dependent pruning. Ablation studies in the supplementary material show that enabling MCF, AGS, and FAC together yields the best performance, whereas removing AGS/FAC or MCF reduces results, and removing all three results in the lowest performance. This supports the notion that the components are non-redundant and most effective when combined.
>
> 5.Does FAC prune fine-grained features? Typically, it does not. FAC is designed based on the observation that the majority of energy in encoder-generated visual features is concentrated in low-frequency components, meaning key features of each frame also lie primarily in low frequencies. PRIM applies a discrete cosine transform (DCT), retains low-frequency blocks, and prunes high-frequency components via a parameter-free module. When applied properly, this effectively removes redundancy without losing critical features.
>
>
> Table X: Efficiency Comparison on WorldSense
> | Backbone | Method | Memory ↓ | Prefilling Time ↓ | Average Accuracy ↑ | Latency per Example ↓ |
> |---|---:|---:|---:|---:|---:|
> | Qwen2.5-Omni-7B | Qwen2.5-Omni | 31G | 289ms (1.00x) | 51.4 | 4.56s (1.00x) |
> | Qwen2.5-Omni-7B | FastV | 57G | 278ms (1.24x) | 44.5 | 4.29s (1.11x) |
> | Qwen2.5-Omni-7B | DyCoke | 31G | 184ms (1.58x) | 44.6 | 3.64s (1.24x) |
> | Qwen2.5-Omni-7B | PRIM (45%) | 27G | 109ms (2.48x) | 50.2 | 3.29s (1.39x) |
> | Qwen2.5-Omni-7B | PRIM (35%) | 25G | 91ms (3.41x) | 47.1 | 3.12s (1.44x) |
> | Qwen2.5-Omni-3B | Qwen2.5-Omni | 28G | 264ms (1.00x) | 51.3 | 3.68s (1.00x) |
> | Qwen2.5-Omni-3B | FastV | 45G | 222ms (1.16x) | 44.4 | 3.45s (1.05x) |
> | Qwen2.5-Omni-3B | DyCoke | 20G | 171ms (1.51x) | 44.0 | 3.12s (1.16x) |
> | Qwen2.5-Omni-3B | PRIM (45%) | 16G | 112ms (2.39x) | 49.9 | 2.92s (1.29x) |
> | Qwen2.5-Omni-3B | PRIM (35%) | 14G | 88ms (3.35x) | 49.4 | 2.79s (1.38x) |

---

> > ### Author Rebuttal · Reviewer_nZxB · 2026-04-01
> >
> > Thank author for clarifying my questions.

---

> > > ### Author Response · Authors · 2026-04-04
> > >
> > > Thank you for your time and for reviewing our clarifications. We sincerely appreciate your thoughtful feedback.

---

### Official Review · Reviewer_QL38 · 2026-03-15

**Soundness:** 2
**Presentation:** 2
**Significance:** 2
**Originality:** 2
**Overall Recommendation:** 3
**Confidence:** 4

**Summary:**

The paper focus on improving inference efficiency in large multimodal models for audio-visual understanding. This manuscript's key aspect comprises PRIM, a cooperative dynamic token compression framework. To address the substantial computational and memory overhead of processing long-form audio-visual content, the method introduces an early multimodal cross fusion module. It then applies an attention-guided selection strategy to prune tokens dynamically, using audio saliency to control video token retention. Additionally, it uses a frequency-aware compaction mechanism to compress visual representations. Finally, a task-aware adaptive pruning module adjusts token retention within the language model based on instruction relevance. Experiments on benchmarks like AVUT, Video-MME, and WorldSense show that the proposed method achieves superior efficiency-accuracy trade-offs.

**Compliance With Llm Reviewing Policy:**

Affirmed.

**Final Justification:**

Overall, I believe this work would benefit from a major revision to reach the quality expected for publication. A revised manuscript may include comprehensive efficiency metrics, cross-architecture verification, and additional baseline comparisons. Given that the initial submission lacks fundamental efficiency comparisons, i am worried incorporating these necessary changes would likely introduce a substantial gap between the reviewed manuscript and the final version.

**Key Questions For Authors:**

see weaknesses

**Limitations:**

yes

**Strengths And Weaknesses:**

Strengths
1. The motivation is clearly presented, with a good analysis of attention distributions to reveal modality imbalance and layer-wise redundancy.
2. The frequency-aware compaction is a simple but effective parameter-free transformation that leverages discrete cosine transform to compress visual features.
3. The cooperative design spanning both multimodal encoders and the language model offers a comprehensive solution rather than just pruning within the language model.


Weaknesses
1. While the method claims to significantly reduce computational and memory overhead , the main tables (Table 1 and Table 2) only report accuracy metrics. There is no direct, comprehensive comparison of floating-point operations, memory usage, or wall-clock time against other efficient baselines like FastV or DyCoke in the main results.
2. Adding the multimodal cross fusion module, which consists of four transformer blocks, introduces additional computational stages before the language model. The overall latency trade-off and the exact overhead of this module need deeper empirical justification.
3. The method introduces several hyperparameters, such as maximum and minimum pruning bounds, retention ratios, and scaling constants. As noted in the supplementary text, performance is highly sensitive to the choice of pruning configurations.
4. The ablation study in Table 3 shows that removing all components drops accuracy by only 1.7% on AVUT and 1.6% on Video-MME, which is surprisingly small. This raises questions about whether the components are truly necessary or whether the baseline is already quite strong. Moreover, the performance of Qwen2.5-Omni and more related baselines  should be added, like Gemini 2.5/3 Pro, Qwen3VL, etc.

---

> ### Author Rebuttal · Authors · 2026-03-30
>
> 1.We agree that the main paper should report latency, FLOPs, and token reduction alongside accuracy. We added comparisons with the uncompressed Qwen2.5-Omni baseline under the same input settings, as well as with FastV and DyCoke under matched token budgets. Tables now report overall retention, FLOPs, GPU memory, prefilling time, and latency.
>
> 2.MCF overhead. The concern is whether the 4-layer MCF offsets token compression savings. Our design concentrates cross-modal interactions in early LLM layers; MCF moves fusion before the backbone rather than adding extra layers, so the backbone processes fused text with compressed audio/video tokens without repeated interactions over long sequences. After fusion and compression, MCF reduces tokens entering the LLM. Fig. 6 shows that FLOPs and latency grow almost linearly with fusion depth, while performance saturates quickly.This is why we chose four layers as the balance point between accuracy and efficiency, rather than adopting deeper layer fusion.
>
> 3. We acknowledge that multimodal token compression is inherently sensitive to pruning configurations, and results in Figure 5 show that overly aggressive pruning of any modality leads to performance drops. However, PRIM is not designed as a rigid fixed-ratio pruning scheme. Instead, the video compression ratio is dynamically allocated based on retained audio saliency within each time window and then normalized to satisfy a fixed global budget. This design reduces the brittleness of manually defined fixed schedules. In supplementary analyses, we further demonstrate that pre-defined pruning ratios are indeed sensitive to configuration choices, whereas our instance-adaptive strategy remains more stable across benchmarks and pruning settings.
>
> 4.We note that Table 3 does not remove the entire PRIM framework; it only removes MCF, AGS, and FAC, while the strong Qwen2.5-Omni backbone and the rest of the pipeline remain unchanged. Therefore, the “all removed” row still represents a strong baseline. In this context, moderate absolute performance drops do not indicate that these components are unnecessary; rather, they show that PRIM successfully preserves accuracy under compression. Finally, we agree that broader baseline coverage would further strengthen the paper’s claims. In the revised version, we will include updated baselines wherever reproducible evaluation is feasible.
>
> | Backbone | Method | Retained Ratio | FLOPs Ratio | IE | CC | EL | CM | OM | TM | AVUT Avg. | Video-MME Avg. |
> |---|---:|---:|---:|---:|---:|---:|---:|---:|---:|---:|---:|
> | Qwen2.5-Omni-7B | Qwen2.5-Omni | 100% | 100% | 78.9 | 41.3 | 51.5 | 75.5 | 67.2 | 65.2 | 63.3 | 77.4 |
> | Qwen2.5-Omni-7B | PRIM (Ours) | 45% | 37% | 78.6 | 39.7 | 51.7 | 74.8 | 64.5 | 64.6 | 62.3 | 77.7 |
> | Qwen2.5-Omni-7B | PRIM (Ours) | 35% | 28% | 77.0 | 39.7 | 47.4 | 70.0 | 62.2 | 64.9 | 60.2 | 77.5 |
> | Qwen2.5-Omni-3B | Qwen2.5-Omni | 100% | 100% | 78.3 | 41.3 | 46.2 | 71.4 | 66.0 | 62.7 | 60.3 | 74.0 |
> | Qwen2.5-Omni-3B | PRIM (Ours) | 45% | 35% | 78.2 | 38.7 | 45.7 | 70.2 | 65.3 | 62.4 | 60.8 | 74.2 |
> | Qwen2.5-Omni-3B | PRIM (Ours) | 35% | 25% | 77.3 | 39.6 | 42.1 | 69.2 | 65.8 | 60.5 | 59.1 | 73.7 |
>
> | Backbone | Method | Retained Ratio | FLOPs (T) | Tech. | Cult. | Daily. | Film. | Performance | Games | Sports | Music | Avg. |
> |---|---:|---:|---:|---:|---:|---:|---:|---:|---:|---:|---:|---:|
> | Qwen2.5-Omni-7B | Qwen2.5-Omni | 100% | 74.2 | 56.7 | 51.4 | 50.6 | 55.8 | 53.3 | 47.9 | 49.8 | 45.5 | 51.4 |
> | Qwen2.5-Omni-7B | PRIM (Ours) | 45% | 29.4 | 54.4 | 52.4 | 49.7 | 55.1 | 49.6 | 47.1 | 50.1 | 44.9 | 50.2 |
> | Qwen2.5-Omni-7B | PRIM (Ours) | 35% | 22.1 | 27.6 | 50.8 | 49.7 | 53.7 | 53.3 | 46.5 | 50.5 | 44.5 | 47.1 |
> | Qwen2.5-Omni-3B | Qwen2.5-Omni | 100% | 39.9 | 55.8 | 52.1 | 47.1 | 56.6 | 53.3 | 48.8 | 52.4 | 44.3 | 51.3 |
> | Qwen2.5-Omni-3B | PRIM (Ours) | 45% | 15.9 | 54.4 | 51.8 | 46.0 | 56.8 | 50.0 | 47.1 | 51.9 | 41.3 | 49.9 |
> | Qwen2.5-Omni-3B | PRIM (Ours) | 35% | 10.7 | 53.1 | 50.2 | 43.9 | 57.6 | 49.3 | 48.8 | 50.8 | 41.3 | 49.4 |
>
> Table X: Efficiency Comparison on WorldSense
> | Backbone | Method | Memory ↓ | Prefilling Time ↓ | Average Accuracy ↑ | Latency per Example ↓ |
> |---|---:|---:|---:|---:|---:|
> | Qwen2.5-Omni-7B | Qwen2.5-Omni | 31G | 289ms (1.00x) | 51.4 | 4.56s (1.00x) |
> | Qwen2.5-Omni-7B | FastV | 57G | 278ms (1.24x) | 44.5 | 4.29s (1.11x) |
> | Qwen2.5-Omni-7B | DyCoke | 31G | 184ms (1.58x) | 44.6 | 3.64s (1.24x) |
> | Qwen2.5-Omni-7B | PRIM (45%) | 27G | 109ms (2.48x) | 50.2 | 3.29s (1.39x) |
> | Qwen2.5-Omni-7B | PRIM (35%) | 25G | 91ms (3.41x) | 47.1 | 3.12s (1.44x) |
> | Qwen2.5-Omni-3B | Qwen2.5-Omni | 28G | 264ms (1.00x) | 51.3 | 3.68s (1.00x) |
> | Qwen2.5-Omni-3B | FastV | 45G | 222ms (1.16x) | 44.4 | 3.45s (1.05x) |
> | Qwen2.5-Omni-3B | DyCoke | 20G | 171ms (1.51x) | 44.0 | 3.12s (1.16x) |
> | Qwen2.5-Omni-3B | PRIM (45%) | 16G | 112ms (2.39x) | 49.9 | 2.92s (1.29x) |
> | Qwen2.5-Omni-3B | PRIM (35%) | 14G | 88ms (3.35x) | 49.4 | 2.79s (1.38x) |

---

> > ### Author Rebuttal · Reviewer_QL38 · 2026-04-05
> >
> > I thank the authors for their rebuttal. While several of my initial concerns have been adequately addressed, but some important concerns remain:
> >
> > W1. Partially resolved. When applying PRISM with the Qwen2.5-Omni-7B backbone at a 35% retention ratio, the performance on the "tech" subset of WorldSense exhibits an anomaly. It undergoes a significant degradation compared to both the 100% and 45% retention ratios, which requires further explanation.
> >
> > W2 & W3. Resolved.
> >
> > W4. Partially resolved or unresolved. First, Table 3 does not substantiate the claim that "PRISM successfully preserves accuracy under compression," as explicit compression metrics are absent. Second, it is unclear what constitutes the "rest of the pipeline" and precisely which components within it influence to the efficiency and accuracy metrics. Finally, verifying the generality of a token compression model across diverse architectural backbones is a standard practice in related literature (e.g., FastV, Xcomp) and remains necessary here.

---

> > > ### Author Response · Authors · 2026-04-05
> > >
> > > 1.Thank you for the careful reading. This issue is due to a typo in the table rather than an actual anomaly. For PRIM with Qwen2.5-Omni-7B at 35% retention on WorldSense, the Tech. score should be 57.6 instead of 27.6; correspondingly, the Avg. should be 50.8 instead of 47.1. We apologize for the mistake and will correct it in the revised version. We sincerely thank the reviewer for catching this error.
> > >
> > > 2.Our PRIM model is a multimodal collaborative token compression framework, in which the four compression modules work together to perform the compression. Therefore, multiple parameters need to be set in the experiments to control different stages. The Retained Ratio refers to the percentage of tokens remaining after compression relative to the original total number of tokens. Here, we set S = 0.3, ρv = 0.6 and S = 0.4, ρv = 0.7 to achieve overall retained ratios of 45% and 35%, respectively, i.e., 55%/65% compression relative to the 100% baseline, while the remaining parameters are kept the same as described in the Implementation Details. Therefore, Table X reports accuracy under concrete compression levels rather than in the absence of compression metrics. Since our implementation is based on Qwen2.5-Omni, the Method entry for Qwen2.5-Omni in the tables X corresponds to the uncompressed model, i.e., a 100% retained ratio. We then use the 3B and 7B versions of Qwen2.5-Omni as the backbone and include two compression strategies (FastV and DyCoke) as baselines to ensure rigorous comparative analysis. FastV ,during its prefill stage, utilizes the attention score matrix of the L-th layer to evaluate token relevance, subsequently pruning tokens. DyCoke represents the first dynamic token compression strategy proposed for VideoLLMs. We employ its first-stage token temporal merging module to process video and audio tokens.
> > >
> > > PRIM consists of the vision encoder, audio encoder, projector, large language model backbone, and our proposed compression module. All four compression modules of PRIM contribute to improved efficiency, primarily by reducing the number of audio/video tokens processed by the large language model. The compression strength of PRIM can affect accuracy: moderate compression removes irrelevant or redundant information, allowing the model to focus on important content, while excessive compression can lead to information loss and decreased model performance. In addition, Table X aims to analyze efficiency under different compression strengths, compression strategies, and backbone models. The ablation effects of individual components are detailed in the main text and supplementary materials.
> > >
> > > We validated our approach on Qwen2.5-Omni models of different scales (7B and 3B). The results show that our method can significantly improve efficiency while maintaining model performance across different model scales, indicating that the approach is robust for varying sizes of the same architecture. Furthermore, our compression method operates on the input tokens and does not rely on any specific visual/audio encoder or the internal attention matrices of the LLM. Therefore, in principle, it can be applied to any full-modality LLM architecture that accepts audio-video tokens, suggesting potential generalizability to other architectures. We agree that validation on more diverse backbone architectures would further strengthen the paper. Our current experiments only cover two scales within the Qwen2.5-Omni family, which supports robustness across model scales but does not fully establish cross-backbone generality. We will make this limitation explicit in the manuscript and leave broader cross-backbone validation as important future work.
> > >
> > > We sincerely thank the reviewer for their time and careful evaluation of our work.

---

### Decision · Program_Chairs · 2026-04-30

**Decision:**

Accept (regular)

**Comment:**

The paper presents PRIM, a cooperative dynamic token compression framework designed to improve the inference efficiency of large multimodal models (LMMs) for audio-visual understanding. Driven by empirical analysis showing modality imbalance (audio tokens receiving higher attention than video) and layer-wise redundancy, PRIM introduces a pipeline comprising early text-conditioned audio-visual fusion (MCF), attention-guided token selection (AGS), frequency-aware compaction via DCT (FAC), and task-aware adaptive pruning inside the LLM.

Following the rebuttal, the reviewers reached a positive consensus. Reviewers nZxB and PLBp explicitly stated their concerns were fully resolved and raised their scores, while Reviewer U2kV maintained a positive score noting their concerns were mostly addressed. Reviewer QL38 acknowledged the improvements but maintained a minor reservation regarding the lack of cross-architecture validation (which the authors acknowledged as a limitation to be explicitly stated).

Given the technical soundness of the proposed framework, the strong empirical motivation, and the thorough rebuttal that successfully addressed the reviewers' primary methodological and evaluation concerns, this paper represents a solid contribution to the efficient multimodal foundation model literature. I recommend it for acceptance if there is room in the program.